

# Kibble–Zurek mechanism in the Ising Field Theory

Kristóf Hódsági[1*] and Marton Kormos[2]

**1** BME-MTA Statistical Field Theory Research Group, Institute of Physics,
Budapest University of Technology and Economics, 1111 Budapest, Budafoki út 8, Hungary
**2** MTA-BME Quantum Dynamics and Correlations Research Group,
Budapest University of Technology and Economics, 1111 Budapest, Budafoki út 8, Hungary

\* hodsagik@phy.bme.hu

## Abstract

The Kibble–Zurek mechanism captures universality when a system is driven through a continuous phase transition. Here we study the dynamical aspect of quantum phase transitions in the Ising Field Theory where the quantum critical point can be crossed in different directions in the two-dimensional coupling space leading to different scaling laws. Using the Truncated Conformal Space Approach, we investigate the microscopic details of the Kibble–Zurek mechanism in terms of instantaneous eigenstates in a genuinely interacting field theory. For different protocols, we demonstrate dynamical scaling in the non-adiabatic time window and provide analytic and numerical evidence for specific scaling properties of various quantities. In particular, we argue that the higher cumulants of the excess heat exhibit universal scaling in generic interacting models for a slow enough ramp.



# 1   Introduction

The Kibble–Zurek mechanism (KZM) describes the dynamical aspects of phase transitions and captures the universal features of nonequilibrium dynamics when a system is driven slowly across a continuous phase transition. The original idea is due to Kibble, who studied cosmological phase transitions in the early Universe [1, 2]. He showed that as the Universe cools below the symmetry breaking temperature, instead of perfect ordering, domains form and topological excitations are created. Not much later Zurek pointed out that this phenomenon can be observed in condensed matter systems as well, and that the density of defects depends on the cooling rate [3, 4]. The physical mechanism originates in the fact that at a critical point both the correlation length and the correlation time (relaxation time) diverge, leading to an inevitable breakdown of adiabaticity. As a consequence, the final state will not be perfectly ordered but will consist of domains with different symmetry breaking orders separated by defects or domain walls. However, in the process a typical time scale and a corresponding length scale emerges related to the instant when the system deviates from the adiabatic course. These quantities, diverging as the rate at which the phase transition is crossed approaches zero, are the only scales in the problem. As a consequence, the density of domain walls as well as other quantities obey scaling laws in terms of the speed of the ramp.

It is a natural question whether the same phenomena occur also at zero temperature, i.e. for quantum critical points. A systematic study of the KZM in quantum phase transitions started with the works [5–8]. Quantum phase transitions are different from transitions at finite temperature: they correspond to a qualitative change in the ground state of a quantum system and are driven by quantum fluctuations. Importantly, the time evolution is unitary and there is no dissipation. In spite of these differences, the scaling behaviour essentially coincides with the classical case [5–10]. The scaling behaviour was extended to other observables beyond the defect density to correlation functions [11–13], entanglement entropy [13–15], excess

heat [16–18], and also to different ramp protocols [10, 16, 19–21], including quenches from the ordered to the disordered phase. The scaling laws can also be derived using the framework of adiabatic perturbation theory [7, 16, 17, 19, 22–25]. The reader interested in the KZM in the context of quantum phase transitions is referred to the excellent reviews [26–28].

The simplest approximation which leads to the right scaling exponents assumes that when adiabaticity is lost, the system becomes completely frozen and reenters the dynamics only some time after crossing the critical point. This freeze-out scenario or impulse approximation has been refined recently by taking into account the actual evolution of the system in the non-adiabatic time window [15, 29–35]. Since the Kibble–Zurek length and time scales are the only relevant scales, the non-adiabatic evolution features dynamical scaling, i.e. the time dependence of various observables is given by scaling functions. This can also be understood from an adiabatic renormalization group approach [20, 21].

The Kibble–Zurek mechanism was also extended beyond the mean values to the full statistics of observables. The number distribution of defects was computed in the Ising chain [13, 36] and was argued to exhibit universality [37]. Similarly, the work statistics and its cumulants were also studied and found to satisfy scaling relations [38–40].

The quantum KZM has been investigated experimentally in cold atomic systems [41–45], including the dynamical scaling [46, 47] and very recently, the number distribution of the defects [48].

The various facets of the quantum KZM were demonstrated and analysed on the quantum Ising chain [6–8, 10, 13, 30, 33, 35, 36, 39, 40, 49–52], the XY spin chain [11, 12, 53] or other exactly solvable systems [15, 31, 50, 54, 54–56] (see however e.g. [9, 18, 34, 57, 58]). Most studies focused on spin chains or other lattice systems, while field theories received less attention. Notable exceptions are Refs. [31, 54–56] and applications of the adiabatic perturbation theory approach to the sine–Gordon model [17, 23, 59]. The KZM in the field theory context also appeared in the context of holography [60–64].

In this work we aim to study different aspects the quantum Kibble–Zurek mechanism in a simple but nontrivial field theory, the paradigmatic Ising Field Theory. This theory is an ideal testing ground as it allows one to study both free and genuinely interacting integrable systems. Our motivation for this choice is twofold. First, we wish to study the KZM in a field theory at the microscopic level of states. Second, we would like to test the recent predictions for the universal dynamical scaling and the scaling behaviour of the higher cumulants of the work in an interacting model.

As we focus on an interacting theory, we need to use a numerical tool for our studies. We use a nonperturbative numerical method, the Truncated Space Approach [65–67]. Apart from its long-standing history to capture equilibrium properties of perturbed conformal field theories [68–80], recent applications demonstrate that it is capable to describe non-equilibrium dynamics in different models [81–86]. This approach gives us access to the microscopic data and full statistics of observables so we can investigate the KZM at work at the lowest level, and being nonperturbative and independent of integrability, it allows us to study the dynamics of the interacting field theory.

The paper is organised as follows. In Sec. 2 we outline the context of our work and review the scaling laws predicted by the Kibble–Zurek mechanism for quantum phase transitions. We proceed by defining the model in which we study the Kibble–Zurek mechanism and discuss the adiabatic perturbation theory that provides another viewpoint on the scaling laws. The main body of the text presents an in-depth analysis of the Kibble–Zurek mechanism in the Ising Field Theory. In Sec. 3 we explore the implications of driving a system across a critical point on the statistics of work function and examine the behaviour of energy eigenstates to check the hypothesis of the KZM at a fundamental level. Sec. 4 discusses the dynamical critical scaling with the time and length scales corresponding to the deviation from the adiabatic course and

demonstrates that the KZ scaling can be observed in the interacting $E_8$ model. In Sec. 5 we show that the appearance of the scaling connected to the Kibble–Zurek mechanism is not limited to local observables but it is present also in higher cumulants of the distribution of the excess heat. Finally, Sec. 6 finishes the paper with concluding remarks and possible future directions. Technical details concerning the relation of the adiabatic perturbation theory to the $E_8$ model, the scaling limit of the analytic solution of the dynamics on the transverse field Ising chain and the applicability of TCSA to the study of KZM are discussed in the Appendices.

## 2 Model and methods

In this section we describe the context of our work by introducing the concepts of the universal non-adiabatic behaviour that manifests itself in power-law dependence of several quantities on the time scale of the non-equilibrium ramp protocol, known under the name of Kibble–Zurek scaling. Then we discuss the model in which we study the KZ scaling, the Ising Field Theory which is the low energy effective theory of the transverse field Ising chain in the vicinity of its critical point. After introducing its main properties, we address the methods that are going to be used to examine the Kibble–Zurek scaling. In the limit of slow ramps, one can employ a perturbative approach, the adiabatic perturbation theory (APT) to investigate the time evolution. We give an overview of this approach, focusing on its application to universal dynamics near quantum critical points. The non-equilibrium dynamics of the Ising Field Theory is amenable to an efficient numerical non-perturbative treatment based on the truncated conformal space approach (TCSA), which we review briefly at the end of the section.

### 2.1 The Kibble–Zurek mechanism

In this section we summarise the KZ scaling laws in a fairly general fashion. Let us consider a perturbation of a quantum critical point (QCP) by some operator with scaling dimension $\Delta$. The strength of the perturbation is characterised by a coupling constant $\delta$ with $\delta = 0$ corresponding to the critical point. Imagine that we prepare the system in its ground state and drive it through its QCP by changing $\delta$ in time, i.e. by performing a ramp. For the sake of generality, we consider ramps that cross the phase transition in a power-like fashion, i.e. near the QCP

$$\delta = \delta_0 \left| \frac{t}{\tau_Q} \right|^a \operatorname{sgn}(t), \qquad (2.1)$$

where $\tau_Q$ is the rate of the quench. The essence of the KZM is that due to the divergence of the relaxation time of the system at the QCP, known as critical slowing down, the system cannot follow adiabatically the change no matter how slow it is, and falls out of equilibrium meaning that it will be in an excited state with respect to the instantaneous Hamiltonian. However, due to universality near the critical point the time and length scales corresponding to the deviation from the adiabatic course depend on the quench rate $\tau_Q$ as a power-law. The scaling can be determined by the following simple argument. The correlation length diverges in the phase transition corresponding to this particular perturbation as $\xi \propto \delta^{-\nu}$ where $\nu$ is the standard equilibrium critical exponent related to the scaling dimension $\Delta$ of the perturbing operator by $\nu = (2 - \Delta)^{-1}$. Similarly, the correlation or relaxation time diverges as $\xi_t \propto \xi^z \propto \delta^{-\nu z}$, where $z$ is the dynamical critical exponent. If the change of $\xi_t$ within a relaxation time is much smaller than the relaxation time itself, $\dot{\xi}_t \xi_t \ll \xi_t$, then the evolution is adiabatic. This is the case for times

$$|t| \gg \tau_{KZ} \equiv (a \nu z)^{\frac{1}{a\nu z+1}} \left( \frac{\tau_Q}{\delta_0^{1/a}} \right)^{\frac{a\nu z}{a\nu z+1}}. \qquad (2.2)$$

However, once we reach $t \approx -\tau_{KZ}$, the rate of change of the correlation time becomes $\dot{\xi}_t \approx 1$ and the evolution becomes non-adiabatic. At this Kibble–Zurek time $\tau_{KZ}$, the correlation time scales with the quench rate $\tau_Q$ as $\tau_{KZ}$ itself:

$$\xi_t(-\tau_{KZ}) \propto \left(\frac{\tau_Q}{\delta_0^{1/a}}\right)^{\frac{a\nu z}{a\nu z+1}} \propto \tau_{KZ}. \qquad (2.3)$$

The first formulation of Kibble–Zurek arguments depicted the non-adiabatic interval of time evolution as a simple freeze-out referring to the assumption that the state is literally frozen in the non-adiabatic regime $t \in [-\tau_{KZ}, \tau_{KZ}]$. At $t = \tau_{KZ}$ on the other side of the QCP, the system finds itself in an excited state with correlation length $\xi_{KZ} = \xi(-\tau_{KZ})$. If the system is now in the ordered phase, it implies that the typical linear size of the ordered domains are $\sim \xi_{KZ}$, so the density of excitations corresponding to defects (domain walls) in spatial dimension $d$ is

$$n_{ex} \propto \xi_{KZ}^{-d} \propto \left(\frac{\tau_Q}{\delta_0^{1/a}}\right)^{-\frac{a\nu d}{a\nu z+1}}. \qquad (2.4)$$

Recently, the freeze-out scenario was refined by taking into account the evolution of the system and change of the correlation length in the time interval $-\tau_{KZ} < t < \tau_{KZ}$ [29–31, 33]. The latter is caused by moving domain walls at the typical velocity corresponding to their typical wave number $k \sim \xi_{KZ}^{-1}$ and energy $\varepsilon(k) \sim k^z \sim \xi_{KZ}^{-z}$. The velocity of this "sonic horizon" [33] is $v = \varepsilon'(k) \sim k^z/k \sim \xi_{KZ}^{1-z}$. The correlation length at $t = \tau_{KZ}$ is then

$$\xi(\tau_{KZ}) = \xi(-\tau_{KZ}) + 2\nu 2\tau_{KZ} = \xi_{KZ}(1 + 4\tau_{KZ}/\xi_{KZ}^z) = \xi_{KZ}(1 + 4\tau_{KZ}/\xi_t(-\tau_{KZ})), \qquad (2.5)$$

which, due to Eq. (2.3), is proportional to $\xi_{KZ}$. This means that $\xi_{KZ}$ is still the only relevant length scale so the scaling laws are not altered.

Still, nontrivial predictions can be made concerning the non-adiabatic or impulse region $-\tau_{KZ} < t < \tau_{KZ}$ [31, 33, 34] due to the fact that the KZ time and correlation length, $\tau_{KZ}$ and $\xi_{KZ}$, are the only relevant scales for a slow enough ramp protocol. Consequently, time-dependent correlation functions are described in terms of scaling functions of the rescaled variables $t/\tau_{KZ}$ and $x/\xi_{KZ}$ in the *KZ scaling limit* $\tau_{KZ} \to \infty$. For example, one- and two-point functions of an operator $\mathcal{O}_{\Delta_\mathcal{O}}$ with scaling dimension $\Delta_O$ take the form in the impulse regime $t \in [-\tau_{KZ}, \tau_{KZ}]$

$$\left\langle \mathcal{O}_{\Delta_\mathcal{O}}(x,t) \right\rangle = \xi_{KZ}^{-\Delta_\mathcal{O}} F_\mathcal{O}(t/\tau_{KZ}), \qquad \left\langle \mathcal{O}_{\Delta_\mathcal{O}}(x,t) \mathcal{O}_{\Delta_\mathcal{O}}(0,t') \right\rangle = \xi_{KZ}^{-2\Delta_\mathcal{O}} G_\mathcal{O}\left(\frac{t-t'}{\tau_{KZ}}, \frac{x}{\xi_{KZ}}\right), \qquad (2.6)$$

where $F$ and $G$ are scaling functions depending on the operator $\mathcal{O}$ and we assumed translational invariance. Note that for one-point functions the scaling holds in the adiabatic regime $t < -\tau_{KZ}$ as well, since there the expectation value depends only on the distance from the critical point, which is the function of the dimensionless time $t/\tau_Q$:

$$\left\langle \mathcal{O}_{\Delta_\mathcal{O}}(x,t) \right\rangle \propto \xi(t)^{-\Delta_\mathcal{O}} \propto \left(\frac{t}{\tau_Q}\right)^{a\nu\Delta_\mathcal{O}} \propto \left(\frac{t}{\tau_{KZ}}\right)^{a\nu\Delta_\mathcal{O}} \tau_{KZ}^{-\Delta_\mathcal{O}/z}, \qquad (2.7)$$

where in the last step we used the relation (2.2).

Considering the generic nature of arguments presented above it is tempting to ask how precisely they describe the actual non-equilibrium dynamics of quantum systems. The scaling relations are supported by exact calculations in the free fermionic Ising chain where the dynamics of low-energy modes can be mapped to the famous Landau–Zener transition problem [5, 8, 33, 87]. In other quantum phase transitions, when exact solutions are not available,

the scaling can be analysed by a perturbative expansion in the derivative of the time-dependent coupling as a small parameter. This approach that uses adiabatic perturbation theory predicts the same scaling as the arguments of Kibble–Zurek mechanism in several models besides the Ising chain [7, 17, 19, 23]. This formalism is useful to apply the generic scaling arguments outside the non-adiabatic regime for quantities that are beyond the scope of the initial formulation of KZM [40]. Together with the non-perturbative numerical method employed in our work it can be used to establish the validity of the scaling relations listed above for an interacting model as well.

To do so, we have to address the question of finite size effects. These are of importance due to the fact that the TCSA method requires finite volume, while the arguments presented above make use of a divergent length scale $\xi_{\mathrm{KZ}}$. Clearly, finite volume can bring about adiabatic behaviour if

$$\xi_{\mathrm{KZ}} \simeq L \quad \Rightarrow \quad \left(\tau_Q/\xi_t\right)^{\frac{a\nu}{a\nu z+1}} \simeq L/\xi, \tag{2.8}$$

where $\xi$ and $\xi_t$ are the correlation length and time at the initial state. If the quench rate $\tau_Q$ is significantly larger than this, the transition is adiabatic due to the fact that finite volume opens the gap. One way to compensate this effect is the rescaling of the volume parameter with the appropriate power of the quench rate [30]. However, if

$$\tau_Q/\xi_t \ll (L/\xi)^{\frac{a\nu z+1}{a\nu}}, \tag{2.9}$$

then the finite size effects are negligible. As we are going to illustrate in Sec. 3.3, we can set the parameters of the numerical TCSA method such that this relation is satisfied and there is no need to rescale the volume parameter.

## 2.2 KZM in the Ising Field Theory

After setting up the context of our work, we now turn to the model in consideration: the Ising Field Theory that is the scaling limit of the critical transverse field Ising chain. The Hamiltonian of the latter reads

$$H_{\mathrm{TFIC}} = -J \left( \sum_i \sigma_i^x \sigma_{i+1}^x + h_x \sum_i \sigma_i^x + h_z \sum_i \sigma_i^z \right), \tag{2.10}$$

where $\sigma_i^\alpha$ with $\alpha = x, y, z$ are the Pauli matrices at site $i$, the strength of the ferromagnetic coupling $J$ sets the energy scale, and $h_x J$ and $h_z J$ are the longitudinal and transverse magnetic fields, respectively. We set periodic boundary conditions, $\sigma_{L+1}^\alpha = \sigma_1^\alpha$. The model is fully solvable in the absence of the longitudinal field, $h_x = 0$, when it can be mapped to free Majorana fermions via the nonlocal Jordan–Wigner transformation. The Hilbert space is composed of two sectors based on the conserved parity of the fermion number. The fermionic Hamiltonian will be local provided we impose anti-periodic boundary conditions for the fermionic operators in the even Neveu–Schwarz (NS) sector and periodic boundary conditions in the odd Ramond (R) sector.

The transverse field Ising model is a paradigm of quantum phase transitions: in infinite volume, for $h_z < 1$ the ground state manifold is doubly degenerate, spontaneous symmetry breaking selects the states $(|0\rangle_{\mathrm{NS}} \pm |0\rangle_{\mathrm{R}})/\sqrt{2}$ with finite magnetisation $\langle\sigma\rangle = \pm(1 - h_z^2)^{1/8}$ (here $|0\rangle_{\mathrm{NS/R}}$ are the ground states in the two sectors). In finite volume, there is an energy split between the states $|0\rangle_{\mathrm{NS}}$ and $|0\rangle_{\mathrm{R}}$ which is exponentially small in the volume, and the ground state is $|0\rangle_{\mathrm{NS}}$. In the paramagnetic phase for $h_z > 1$, the ground state is always $|0\rangle_{\mathrm{NS}}$ and the magnetisation vanishes. The quantum critical point (QCP) separating the ordered and disordered phases is located at $h_z = 1$, which can also be seen from the behaviour of the gap, $\Delta = 2J|1 - h_z|$, vanishing at the QCP. In the ferromagnetic phase, the massive fermionic

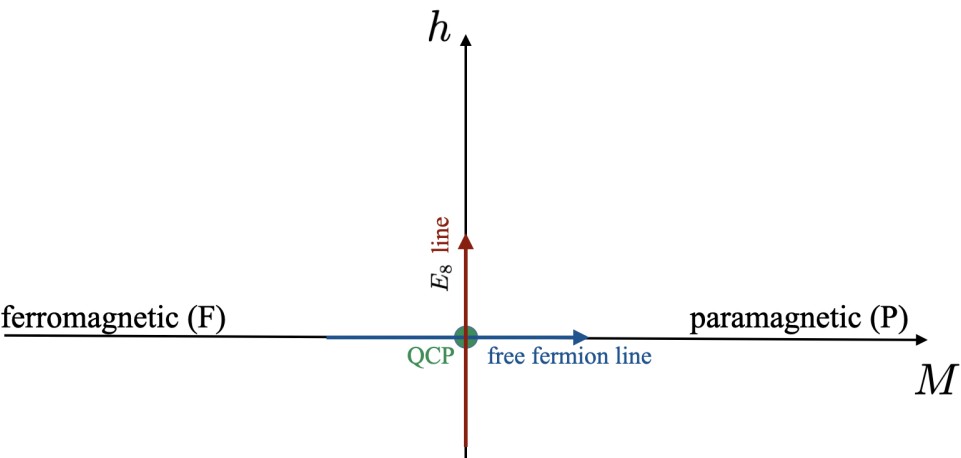

Figure 2.1: Phase diagram of the Ising Field Theory. The couplings $M$ and $h$ characterise the strengths of the perturbations of the $c = 1/2$ conformal field theory by its two relevant operators, $\varepsilon$ and $\sigma$. The KZM is studied for ramps along the integrable directions indicated by the coloured arrows.

excitations can be thought of AS domain walls separating domains of opposite magnetisations, and with periodic boundary conditions their number is always even [1]. In the paramagnetic phase the excitations are essentially spin flips in the $z$ direction.

For $h_x \neq 0$ the model is not integrable[2] for any value of $h_z$, but features weak confinement: the nonzero longitudinal field splits the degeneracy between the two ground states with an energy difference proportional to the system size. The domain walls cease to be freely propagating excitations, as the energy cost increases with the distance between two neighbouring domain walls that have a domain of the wrong magnetisation between them. The new excitations are a tower of bound states, sometimes called 'mesons' in analogy with quark confinement in the strong interaction.

The low energy effective theory describing the model near the critical point is the Ising field theory, obtained in the scaling limit $J \to \infty$, $a \to 0$, $h_z \to 1$ such that speed of light $c_\ell = 2Ja$ and the gap $\Delta = 2J|1 - h_z|$ are fixed ($a$ is the lattice spacing). The critical point corresponds to the theory of a free massless Majorana fermion, which is also one of the simplest conformal field theories (CFT). Due to relativistic invariance, the dynamical critical exponent is $z = 1$. The two relevant operators at the quantum critical point are the magnetisation $\sigma$ (scaling dimension $1/8$) and the so-called 'energy density' $\varepsilon$ (scaling dimension 1), corresponding to the longitudinal and transverse magnetic fields in the scaling limit. The Hamiltonian of the resulting field theory in finite volume $L$ is given by

$$H_{\mathrm{IFT}} = H_{\mathrm{FF}} + \frac{M}{2\pi} \int_0^L \varepsilon(x)\mathrm{d}x + h \int_0^L \sigma(x)\mathrm{d}x. \tag{2.11}$$

Here $H_{\mathrm{FF}}$ is the Hamiltonian of the free massless Majorana fermion, a minimal CFT with central charge $c = 1/2$. The precise relations between the lattice and continuum versions of the

---

[1]This is true even in the Ramond sector, as $|0\rangle_{\mathrm{R}}$ contains a zero-momentum particle.

[2]The $\sigma_i^x$ operators are nonlocal in terms of the fermions so the Jordan–Wigner transformation does not lead to a local fermionic Hamiltonian.

longitudinal magnetic field and the magnetization operator are

$$\sigma(x = ja) = \bar{s} J^{1/8} \sigma_j^x \, , \tag{2.12}$$

$$h = 2\bar{s}^{-1} J^{15/8} h_x \, , \tag{2.13}$$

with $\bar{s} = 2^{1/12} e^{-1/8} \mathcal{A}^{3/2}$ where $\mathcal{A} = 1.2824271291\ldots$ is Glaisher's constant.

For $h = 0$ the Hamiltonian describes the dynamics of a free Majorana fermionic field with mass $|M|$ (we set the speed of light to one, $c_\ell = 1$). We will refer to this choice of parameters in the $M-h$ parameter plane of the theory (2.11) as the "*free fermion line*" (see Fig. 2.1). The QCP at $M = 0$ separates the paramagnetic phase $M > 0$ from the ferromagnetic phase $M < 0$. The coupling is proportional to the mass gap and since the correlation length is the inverse of the gap, $\nu = 1$.

Interestingly, there is another set of parameters that corresponds to an integrable field theory: $M = 0$ with $h$ finite[3]. The spectrum of this theory can be described in terms of eight stable particles, the mass ratios and scattering matrices of which can be written in terms of the representations of the exceptional $E_8$ Lie group. From now on, we are going to refer to this specific set of parameters as the "$E_8$ *integrable line*" (see Fig. 2.1). The lightest particle with mass $m_1$ sets the energy scale which is connected to the coupling $h$ as

$$m_1 = (4.40490857\ldots)|h|^{8/15} \, . \tag{2.14}$$

The exponent reflects that along the $E_8$ line ($\sigma$ perturbation) $\nu = 8/15$ and $z = 1$. Moving particle states are built up as combinations of particles with finite momenta from the same or different species. The masses of the these particle species can be expressed in terms of $m_1$ as [88]

$$m_2 = 2m_1 \cos\frac{\pi}{5} \, , \quad m_3 = 2m_1 \cos\frac{\pi}{30} \, , \quad m_4 = 2m_2 \cos\frac{7\pi}{30} \, , \quad m_5 = 2m_2 \cos\frac{2\pi}{15} \, ,$$

$$m_6 = 2m_2 \cos\frac{\pi}{30} \, , \quad m_7 = 4m_2 \cos\frac{\pi}{5}\cos\frac{7\pi}{30} \, , \quad m_8 = 4m_2 \cos\frac{\pi}{5}\cos\frac{2\pi}{15} \, . \tag{2.15}$$

The exact relation between $m_1$ and the coupling constant $h$ was derived in Ref. [89]. Due to the integrability of the model, the scattering $S$-matrix involving all different species is also known exactly [88]. This, in particular, allows for the identification of multiparticle eigenstates in finite volume based on the volume dependence of their energy.

In the following we are going to consider ramp protocols along the integrable lines, indicated by the coloured arrows in Fig. 2.1. Using the terminology of Ref. [31], we distinguish protocols with $\lambda_i$ and $\lambda_f$ corresponding to different phases of the model (ramp crossing the critical point), and protocols with $\lambda_f = 0$ (ramp ending at the critical point). We are going to refer to these two choices as trans-critical protocol (TCP) and end-critical protocol (ECP), respectively. Certain observables exhibit markedly different behaviour depending on the protocol [40], hence both of them are of interest.

We focus our attention on linear ramps, where one of the couplings is varied such that the system reaches or crosses the critical point at a constant rate,

$$\lambda(t) = -\lambda_0 \frac{t}{\tau_Q} \, , \tag{2.16}$$

where $\lambda$ stands for $M$ or $h$ and the other coupling is set to zero. $\tau_Q$ is the duration of the ramp that takes place in the time interval $t \in [-\tau_Q/2, \tau_Q/2]$ for a TCP ramp and $t \in [-\tau_Q, 0]$ for an ECP ramp.

---

[3]The lattice model is *not* integrable for $h_z = 1$ and $h_x \neq 0$, this is a feature of the field theory in the scaling limit.

Ramps along the free fermion line ($h = 0$) have been studied extensively, especially in the spin chain. The time evolution of the free fermion modes with different momentum magnitudes decouple and only modes of opposite momenta $\{k, -k\}$ are coupled by the evolution equation. One can make progress either by invoking the Landau–Zener description of transitions between energy levels or by numerically solving the set of two differential equations. Even analytical solutions are known for various ramp profiles [26, 54]. These solutions can be simply generalised to the continuum field theory, providing us with an analytical tool to examine the KZ scaling and offering a benchmark for our numerical method. We refer the reader to Appendix B for the details.

The Kibble–Zurek mechanism is much less studied along the other integrable axis $M = 0$. As we noted above, in this direction $\nu = 8/15$, so the KZ scaling is modified with respect to the well-investigated free fermion case. Although the model is integrable, the time evolution cannot be solved analytically, which highlights the importance of the non-perturbative numerical method that exploits the conformal symmetry of the critical model: the Truncated Conformal Space Approach (TCSA). Nevertheless, standard KZ arguments rely only on typical energy and distance scales of the model, consequently they should apply regardless of the presence of interactions. The scaling arguments can be supported by the analysis of the exactly known form factors of the model in the context of the adiabatic perturbation theory, to which we turn now.

## 2.3   Adiabatic Perturbation Theory

The adiabatic perturbation theory (APT) is a standard approach to study the response to a slow perturbation [27, 90]. It was first used to describe the universal dynamics of extended quantum systems in the vicinity of a quantum critical point in Ref. [7]. Ever since the framework has become more elaborate by exploring the parallelism between APT and the Kibble–Zurek mechanism and generalising the arguments to a wider variety of scaling quantities in different models [16, 17, 19, 23–25, 40]. In particular, it has already been applied with success in an integrable field theory, the sine–Gordon model [17]. In our current work we carry out an analogous reasoning to explore the implications of the APT statements in the $E_8$ Ising Field Theory. To this end, let us briefly sketch the basic concepts and assumptions underlying the framework of adiabatic perturbation theory as well as introduce some notations. Our discussion is based on the presentation of Ref. [24].

Assume that we want to solve the time-dependent Schrödinger equation:

$$\imath \frac{\mathrm{d}}{\mathrm{d}t} |\Psi(t)\rangle = H(t) |\Psi(t)\rangle \tag{2.17}$$

in a time interval $t \in [t_\mathrm{i}, t_\mathrm{f}]$. Using the basis of eigenstates of $H(t)$ that are going to be called instantaneous eigenstates $|n(t)\rangle$,

$$H(t) |n(t)\rangle = E_n(t) |n(t)\rangle , \tag{2.18}$$

we can expand the time evolved state with coefficients $\alpha_n(t)$:

$$|\Psi(t)\rangle = \sum_n \alpha_n(t) \exp\{-\imath \Theta_n(t)\} |n(t)\rangle , \tag{2.19}$$

where the dynamical phase factor $\Theta_n(t) = \int_{t_\mathrm{i}}^{t} E_n(t') \, \mathrm{d}t'$ is already included. The initial condition is that at $t_\mathrm{i}$ the system is in its ground state $|0(t_\mathrm{i})\rangle$. Substituting this Ansatz into Eq. (2.17) yields a system of coupled differential equations for the coefficients $\alpha_n(t)$. The resulting system of equations can be solved approximately for $\alpha_n(t)$ using a few assumptions. First, the explicit time-dependence of the Hamiltonian is due to a time-dependent coupling constant $\lambda$ that couples to some perturbing operator $V$ so $H(t) = H_0 + \lambda(t)V$. Second, $\lambda(t)$ is

a monotonous function of time, hence one can perform a change of variables, and it changes slowly (that is the adiabatic assumption) such that $\dot\lambda \to 0$. Then the resulting expression can be expanded in terms of powers of $\dot\lambda$. Assuming there is no Berry phase, the result up to leading order in $\dot\lambda$ is

$$\alpha_n(\lambda) \approx \int_{\lambda_{\mathrm{i}}}^{\lambda} \mathrm{d}\lambda' \left\langle n(\lambda')\middle| \partial_{\lambda'} \middle| 0(\lambda')\right\rangle \exp\left\{\iota(\Theta_n(\lambda') - \Theta_0(\lambda'))\right\}, \tag{2.20}$$

where the dynamical phase with respect to the coupling is $\Theta_n(\lambda) = \int^{\lambda} E_n(\lambda')/\dot\lambda' \, \mathrm{d}\lambda'$ with $\lambda = \lambda(t)$. Note that the phase factor exhibits rapid oscillations in the limit $\dot\lambda \to 0$. This can be exploited to identify the two possibly dominant contributions of integral Eq. (2.20) in this limit. First, a non-analytic part that comes from the saddle point of the phase factor at a complex value of coupling $\lambda$. It is exponentially suppressed with the inverse of the rate $\dot\lambda$. Second, there are contributions coming from the boundaries of the integration domain which can be obtained by integrating by parts and keeping terms to first order in $\dot\lambda$ yields the result

$$\alpha_n(\lambda_{\mathrm{f}}) \approx \iota\dot\lambda' \frac{\left\langle n(\lambda')\middle| \partial_{\lambda'} \middle| 0(\lambda')\right\rangle}{E_n(\lambda') - E_0(\lambda')} \exp\left\{\iota(\Theta_n(\lambda') - \Theta_0(\lambda'))\right\}\bigg|_{\lambda_{\mathrm{i}}}^{\lambda_{\mathrm{f}}}. \tag{2.21}$$

This contribution can be viewed as a switch on/off effect as it is the consequence of a non-smooth start or end of the ramp: it is nonzero if the first time derivative of the coupling has a discontinuity at the initial or final times. If $\dot\lambda_{\mathrm{i,f}} = 0$ then one has to go to higher orders. In general, a discontinuity in the $a$th derivative brings about the scaling $\alpha \propto \tau_{\mathrm{Q}}^{-a}$ with the time parameter of the ramp $\tau_{\mathrm{Q}}$ [26]. We consider linear ramps (cf. Eq. (2.16)) so higher derivatives disappear and the small parameter of the perturbative expansion is $1/\tau_{\mathrm{Q}}$. We remark that Eq. (2.21) can be modified if the energy difference in the denominator vanishes at some time instant along the process, in that case the dependence of $\alpha$ on $\dot\lambda$ is subject to change (cf. Eq. (2.26) for low-momentum modes if the gap is closed).

The applicability of adiabatic perturbation theory, strictly speaking, requires that the overlap between the time-evolved state and the instantaneous ground state remains close to 1 [90]. This, however, imposes a constraint on the probability to be in an excited state rather than on the density of excitations. On the other hand, for quantum many-body systems in the thermodynamic limit the physical criterion for a perturbative treatment is to be in a low-density state [19]. Given that the Kibble–Zurek mechanism predicts that densities decay as a power law of the time parameter $\tau_{\mathrm{Q}}$, in the limit $\tau_{\mathrm{Q}} \to \infty$ the above approximations are justified and we can use Eq. (2.20) to examine the Kibble–Zurek scaling. This reasoning predicts the correct scaling exponents in the transverse field Ising chain for various quantities [24, 40]. Let us illustrate how they work in the case of the density of defects $n_{\mathrm{ex}}$ after a linear ramp $\lambda(t) = \lambda_{\mathrm{i}} + (\lambda_{\mathrm{f}} - \lambda_{\mathrm{i}})t/\tau_{\mathrm{Q}}$. The states of the Ising chain participating in the dynamics are products of zero-momentum particle pair states with momentum $k$, hence the defect density can be expressed as[4]

$$n_{\mathrm{ex}} = \lim_{L \to \infty} \frac{2}{L} \sum_{k>0} |\alpha_k|^2 = \int_{-\pi}^{\pi} \frac{\mathrm{d}k}{2\pi} |\alpha_k|^2, \tag{2.22}$$

where $\alpha_k = \alpha_k(\lambda_{\mathrm{f}})$ is the coefficient of a particle pair state $|k, -k\rangle$ given by Eq. (2.20). To investigate the dependence on $\tau_{\mathrm{Q}}$ it is practical to introduce the rescaled variables

$$\eta = k\tau_{\mathrm{Q}}^{\frac{\nu}{1+z\nu}}, \qquad \zeta = \lambda\tau_{\mathrm{Q}}^{\frac{1}{1+z\nu}}, \tag{2.23}$$

---

[4]We remark that in principle the normalization of the state should be taken into account, but it is 1 up to first order in the perturbation theory.

to remove the $1/\tau_Q$ dependence from the exponent of Eq. (2.20). The heart of the APT treatment of KZ scaling lies at the observation that the matrix element and energy difference appearing in the expression of $\alpha_n$ take the following scaling forms:

$$E_k(\lambda) - E_0(\lambda) = |\lambda|^{z\nu} F(k/|\lambda|^\nu) \tag{2.24}$$

$$\langle \{k, -k\}(\lambda)| \, \partial_\lambda |0(\lambda)\rangle = \lambda^{-1} G(k/|\lambda|^\nu), \tag{2.25}$$

with the asymptotic behaviour $F(x) \propto x^z$ and $G(x) \propto x^{-1/\nu}$ as $x \to \infty$. These considerations yield that

$$n_{\text{ex}} = \tau_Q^{-\frac{\nu}{1+z\nu}} \int \frac{\mathrm{d}\eta}{2\pi} K(\eta), \tag{2.26}$$

with

$$K(\eta) = \left| \int_{\zeta_i}^{\zeta_f} \mathrm{d}\zeta \, \frac{G(\eta/\zeta^\nu)}{\zeta} \exp\left( \iota \int_{\zeta_i}^{\zeta} \mathrm{d}\zeta' \, \zeta'^{z\nu} F(\eta/\zeta'^\nu) \right) \right|^2. \tag{2.27}$$

Eq. (2.26) is analysed in the limit $\tau_Q \to \infty$. In that case the limits of the integral over $\eta$ are sent to $\pm\infty$ and one has to check whether the resulting integral converges or not. Substituting Eqs. (2.24) and (2.25) one can perform the integral in (2.27) in the limit $\eta \gg \zeta_{i,f}^\nu$ to determine the asymptotic behaviour

$$K(\eta) \propto \eta^\beta \equiv \eta^{-2z-2/\nu}. \tag{2.28}$$

The criterion for convergence then is $2z + 2/\nu > 1$, or, equivalently $\frac{\nu}{1+z\nu} < 2$ [24]. In the opposite case the integral is divergent, indicating that to discard the contribution from high-energy modes in the limit $\tau_Q \to \infty$ is not justified. The scaling brought about by all energy scales is quadratic $\tau_Q^{-2}$ due to the discontinuity of $\dot\lambda$, cf. Eq. (2.21). Consequently, the case of equality $\frac{\nu}{1+z\nu} = 2$ distinguishes between the Kibble–Zurek scaling determined by the exponent of $\tau_Q$ in Eq. (2.26) and the quadratic scaling.

### 2.3.1 Application to the Ising Field Theory

These are the key themes of adiabatic perturbation theory as applied to model the Kibble–Zurek mechanism. Now we are going to show that these considerations can be generalised to the two integrable directions of the Ising Field Theory. In the case of the free field theory the generalisation of the arguments above is straightforward and it yields the same result as for the free fermion Ising chain. To apply the reasoning to the $E_8$ integrable model requires a bit of extra work. The complications are mainly technical, details are presented in Appendix A. Here we would like to highlight the key assumptions of the arguments only.

There are several major differences between the free fermion and the $E_8$ field theory: the spectrum of the latter exhibits eight stable stationary particles, moving particle states are built up by combining particles of various species. As a result, there are multiple kinds of many-particle states in contrast to the pair of a single particle species in the free field theory. Interactions between particles modify the simple $p_n = 2\pi n/L$ quantisation rule of momenta in finite volume $L$, leading to a nontrivial density of states in momentum space. Eigenstates of the theory are asymptotic scattering states labelled by the relativistic rapidity variable $\vartheta$ that is related to the energy and momentum of particle $j$ as $E_j = m_j \cosh\vartheta_j$ and $p_j = m_j \sinh\vartheta_j$.

To investigate the Kibble–Zurek scaling in this model we make several simplifying assumptions. First, we consider low-density states which is justified in the limit $\tau_Q \to \infty$. Apart from being a necessary assumption to use the framework of adiabatic perturbation theory, it sets the ground for our second assumption: that is, we assume that the contribution from one- and two-particle states contribute dominantly to intensive quantities such as the defect and energy density. In contrast to the free fermion case, the time-evolved state in the $E_8$ model includes

contributions from multiparticle states that do not factorise exactly to a product of particle pairs. On the other hand, the many-particle overlap functions still satisfy the pair factorisation up to a very good approximation given that the energy density of the non-equilibrium state is low [82, 91] compared to the natural scale set by the final mass gap. Intuitively, the essence of this approximation is that due to large interparticle distance, the interactions between particles located far from each other can be neglected. Hence, the contribution of genuine multiparticle states is proportional to the probability of more than two particles located within a volume related to the correlation length. For a low-density state this probability is indeed tiny, hence the pair factorisation is a good approximation. This assumption is also verified by previous works modeling the non-equilibrium dynamics of the Ising Field Theory that show that time evolution after sudden quenches is dominated by few-particle overlaps in the regime of low post-quench density [81, 84, 92].

Based on these assumptions, we can show that the arguments of APT generalise to an interacting field theory as well. Let us sketch the derivation for the excess heat density $w$ that can be expressed as

$$w(\lambda_f) = \lim_{L \to \infty} \frac{1}{L} \sum_n E_n(\lambda_f) |\alpha_n(\lambda_f)|^2 \,. \tag{2.29}$$

We evaluate this expression by calculating the $\alpha_n$ coefficients as given by Eq. (2.20) in finite volume and then take the $L \to \infty$ limit. Taking into account the finite volume expression of matrix elements in the $E_8$ model, we find that one-particle states contribute to the energy density with the right KZ exponent $\tau_Q^{-\frac{\nu}{\nu+1}}$ (for details see Appendix A.1). To the best of our knowledge, this is the first case when the KZ scaling of one-particle states is investigated in adiabatic perturbation theory.

The contribution of a two-particle state with species $a$ and $b$ is going to be denoted $w_{ab}$ and reads

$$w_{ab}(\lambda_f) = \frac{1}{L} \sum_\vartheta (m_a \cosh\vartheta + m_b \cosh\vartheta_{ab}) |\alpha_\vartheta(\lambda_f)|^2 \,, \tag{2.30}$$

where $\vartheta_{ab}$ is a function of $\vartheta$ determined by the constraint that the state has zero overall momentum. To take the thermodynamic limit one has to convert the summation to an integral over rapidities. The key observation to proceed is that the effects of the interactions are of $\mathcal{O}(1/L)$ and disappear in the limit $L \to \infty$. Consequently, one can change the integration variable such that it goes over momentum instead of the rapidity. From then on, the derivation is identical to the free fermion case, although one has to check whether the scaling forms (2.24) and (2.25) apply for the dispersion and matrix elements of the $E_8$ theory as well. Observing that $\vartheta = \text{arcsinh}(p/m_a) = \text{arcsinh}[p/(c|\lambda|^\nu)]$ with some constant $c$, one can see that the former is trivially satisfied with the right asymptotic $F(x) \propto x^z$. The latter equation regarding the scaling and the high-energy behaviour of the matrix element also holds in general, as one can verify in the $E_8$ model (see Appendix A). Hence, as long as the initial assumptions of low energy and approximate pair factorisation are valid, the adiabatic perturbation theory predicts KZ scaling of intensive quantities in the $E_8$ theory as well.

## 2.4 Truncated Conformal Space Approach

After introducing the perturbative approach to model the scaling laws of the Kibble–Zurek mechanism in the Ising Field Theory, let us now address a non-perturbative numerical method that can be used to verify the arguments above by explicitly simulating the dynamics. In the following we turn our focus to the Truncated Conformal Space Approach and discuss the underlying principles and its operation.

Numerical methods that are based on truncating the Hilbert space have a long history of capturing equilibrium properties of field theories (see [67] for a review). In particular, two-

dimensional field theoretical models that are defined by perturbing a conformal field theory or free theory by relevant operators are amenable to a very efficient numerical treatment, called the Truncated Conformal Space Approach (TCSA) [65,66]. The essential idea of the method is to compute the matrix elements of the perturbing operators in the basis of the unperturbed theory in finite volume where the spectrum is discrete. The resulting Hamiltonian matrix is then made finite dimensional by truncating the basis, hence the name of the method. Recently, it has been applied with success to model the non-equilibrium dynamics of different theories, in particular the Ising Field Theory [81,84,86,92]. We dedicate this section to briefly introduce the method and set up some notation along the course.

To model the Kibble–Zurek mechanism in the Ising Field Theory we define the theory in a finite volume $L$ using periodic boundary conditions, so the space-time covers an infinite cylinder of circumference $L$. The basis states used by TCSA are the energy eigenstates of the $c = 1/2$ free Majorana conformal field theory on the cylinder. The truncation keeps only a finite set of states that diagonalise the conformal Hamiltonian $H_0$ by discarding those with energy larger than a given cut-off $E_{\text{cut}}$. The exact finite volume matrix elements of the primary fields $\sigma$ and $\varepsilon$ can be constructed on this basis by mapping the cylinder to the complex plane where conformal Ward identities can be utilised. Perturbing the CFT opens a mass gap $\Delta$ that can be used to express the Hamiltonian matrix $H$ in a dimensionless form for numerical calculations:

$$H/\Delta = (H_0 + H_\phi)/\Delta = \frac{2\pi}{l}\left(L_0 + \bar{L}_0 - c/12 + \tilde{\kappa}\frac{l^{2-\Delta_\phi}}{(2\pi)^{1-\Delta_\phi}}M_\phi\right), \qquad (2.31)$$

where $l = \Delta L$ is the dimensionless volume parameter, $M_\phi$ is the matrix of the operator $\phi = \sigma, \varepsilon$ having scaling dimension $\Delta_\phi$ with $\Delta_\sigma = 1/8$ and $\Delta_\varepsilon = 1$. Here $\tilde{\kappa}$ is the dimensionless coupling constant that characterises the strength of the perturbation. The ramping protocol is thus realised in TCSA by tuning $\tilde{\kappa}$ linearly in the dimensionless time $\Delta_{\text{i}}t$, where $\Delta_{\text{i}}$ is the mass gap at the initial time instant. All quantities are measured in appropriate powers of $\Delta_{\text{i}}$ along the course of the ramp. Referring to the different physical content of the theories that result from the choice of $\sigma$ or $\varepsilon$ we use different notation for the mass gap in this work. The $\sigma$ perturbation yields the $E_8$ spectrum with eight stable particles hence the notation for the mass gap in this case is $m_1$, the mass of the lightest particle. The $\varepsilon$ direction corresponds to a free fermion field theory with a single species so we simply denote $\Delta$ as $m$ the mass of the elementary excitation.

The success of TCSA to model the physical theory without an energy cut-off relies on its capability to suppress truncation errors as much as possible. Achieving higher and higher cut-offs is computationally demanding but the contribution of high energy states can be taken into account through a renormalisation group (RG) approach [75,79,93–97]. The RG analysis predicts a power-law dependence on the cut-off. Here we use a simpler extrapolation scheme using the powers predicted by the RG analysis which improves substantially the results obtained using relatively low cut-off energies. We express the recipe for extrapolation in terms of the conformal cut-off level $N_{\text{cut}}$ that is related to the energy cut-off as $N_{\text{cut}} = L/(2\pi)E_{\text{cut}}$. One can show that the results for some arbitrary quantity $\phi$ at infinite cut-off are related to TCSA data as

$$\langle\phi\rangle = \langle\phi\rangle_{\text{TCSA}} + AN_{\text{cut}}^{-\alpha_\phi} + BN_{\text{cut}}^{-\beta_\phi} + \dots, \qquad (2.32)$$

where the $\alpha_\phi < \beta_\phi$ exponents are positive numbers depending on the scaling dimension of the perturbation, the operator in consideration and those appearing in their operator product expansion. Ellipses denote further subleading corrections that decay faster as $N_{\text{cut}} \to \infty$. The details of the extrapolation in various cases are detailed in Appendix C.

With this we have finished reviewing the basic concepts in the Kibble–Zurek mechanism and in the Ising Field Theory. We have introduced the two main methods that we use to study

it: the numerical method of TCSA for simulating the dynamics and the scaling arguments in the context of APT that predicts that for the KZ scaling the presence of interactions in the $E_8$ theory makes no difference. We have outlined the following claims: the scaling behaviour observed on the transverse field Ising chain does not change in the continuum limit and that the only modification needed for the interacting $E_8$ model is to take into account the different scaling exponent $\nu$. Before putting these claims to test by calculating the dynamics of one-point functions and observing the statistics of excess heat, we investigate the dynamics of energy eigenstates along the ramp in order to sketch an intuitive picture of how the Kibble–Zurek mechanism can be understood at the most fundamental level.

# 3 Work statistics and overlaps

We aim to study the evolution of the quantum state during the ramp, including the non-adiabatic regime, in detail. Using the TCSA method, we have access to microscopic data, which allows us to investigate the details of the dynamics. There are many possible quantities to consider: the correlation length, excitation densities, etc. In this section we adopt another, more microscopic perspective: we observe how instantaneous eigenstates get populated in the course of the ramp, how the adiabatic behaviour breaks down and how excitations are created. Looking at the fundamental components that conspire to create the well-known KZ scaling in a wide variety of quantities provides us with an intuitive and visual picture about what happens during the regime when adiabaticity is lost.

To this end, we first solve the time-dependent Schrödinger equation:

$$\iota \frac{\mathrm{d}}{\mathrm{d}t} |\Psi(t)\rangle = H(t) |\Psi(t)\rangle \,, \tag{3.1}$$

in the time interval $t \in [-\tau_Q/2, \, \tau_Q/2]$ with the initial state $|\Psi_0\rangle$ chosen to be the ground state of the initial Hamiltonian $H(-\tau_Q/2)$. Since momentum is conserved all along the ramp and the initial state is a zero-momentum state, $|\Psi(t)\rangle$ is also a $P = 0$ state for all $t$.

To characterise how the energy eigenstates get populated we can generalise the statistics of work function [98] to each time instance along the course of the ramp, defining an instantaneous statistics of work function

$$P(\tilde{W}, t) = \sum_n \delta\left(\tilde{W} - [E_n(t) - E_0(0)]\right) |g_n(t)|^2 \,, \tag{3.2}$$

where the sum is running over the spectrum of the instantaneous Hamiltonian $H(t)$ with eigenvalues $E_n(t)$ and eigenstates $|n(t)\rangle$. Here $g_n(t)$ are the overlaps of the time-evolved state with the instantaneous eigenstates:

$$g_n(t) = \langle n(t)|\Psi(t)\rangle \,. \tag{3.3}$$

$\tilde{W}$ is called to the total work performed by the non-equilibrium protocol. $P(\tilde{W}, t)$ is non-zero only if $\tilde{W} \geq E_0(t) - E_0(0)$. In the following we focus only on the statistics of the excess work $W = \tilde{W} - [E_0(t) - E_0(0)]$ so $P(W, t)$ is non-zero if $W \geq 0$.

In order to draw a clear picture of what happens for ramps within the reach of KZM, we present the two sections of $P(W, t)$: first, only the $|g_n(t)|^2$ overlap amplitudes with respect to time and second, the snapshot of $P(W, t)$ at some time instant $t$.

## 3.1 Ramps along the free fermion line

Let us start with the exactly solvable dynamics, i.e. the free fermion line of the model (2.11) corresponding to $h = 0$. The time-dependent coupling is the free fermion mass, $\lambda(t) = M(t)$.

Our ramp protocol is a simple linear ramp profile that is symmetric around the critical point:

$$M(t) = -2M_{\mathrm{i}}t/\tau_{\mathrm{Q}}, \qquad (3.4)$$

where $M_{\mathrm{i}}$ is the initial value of the coupling at $t = -\tau_{\mathrm{Q}}/2$. As discussed in Sec. 2.2, the critical exponents in this case are $\nu = 1$, $z = 1$, so the Kibble–Zurek time (2.2) scales as $\tau_{\mathrm{KZ}} \sim \sqrt{\tau_{\mathrm{Q}}}$. For testing the various scaling forms we need to have a specified value of $\tau_{\mathrm{KZ}}$ which we simply set as

$$m\tau_{\mathrm{KZ}} = \sqrt{m\tau_{\mathrm{Q}}}, \qquad (3.5)$$

where $m = |M_{\mathrm{i}}|$ is the mass gap at the start of the ramp. Depending on the sign of $M_{\mathrm{i}}$, the ramp is either towards the ferromagnetic phase or the paramagnetic phase; we are going to present our results in this order.

### 3.1.1 The paramagnetic-ferromagnetic (PF) direction

Ramps starting from the paramagnetic phase are defined by $M_{\mathrm{i}} > 0$. In this case the ground state is non-degenerate and lies in the Neveu–Schwarz sector, so the time evolved state is orthogonal to the Ramond sector subspace for all times (see Sec. 2.2).

Analogously to the lattice dynamics, starting from the ground state at a given $M_{\mathrm{i}}$, only states consisting of zero-momentum particle pairs have nonzero overlap with the time evolved state, moreover, the different pairs of momentum modes $\{p, -p\}$ decouple completely. In finite volume $L$ the momentum is quantised as $p_n = 2\pi n/L$, where $n$ is half-integer in the NS sector. To solve the dynamics we follow the approach of [54] and use the Ansatz:

$$|\Psi(t)\rangle = \bigotimes_p |\Psi(t)\rangle_p, \qquad \text{with} \qquad |\Psi(t)\rangle_p = a_p(t)|0\rangle_{p,t} + b_p(t)|1\rangle_{p,t}, \qquad (3.6)$$

where $|0\rangle_{p,t}$ and $|1\rangle_{p,t}$ denote the instantaneous ground and excited states of the two-level system at time $t$ along the ramp. The coefficients $a_p(t)$ and $b_p(t)$ satisfy $|a_p(t)|^2 + |b_p(t)|^2 = 1$ and they can be expressed via the solutions of two coupled first order differential equations (for details see the Appendix B). The population of mode $p$ is given by $n_p(t) = |b_p(t)|^2$. Although the equations can be solved exactly, numerical integration is more suitable for our purposes. Hence, strictly speaking, referring to this solution as 'analytical' is not entirely precise. From now on, when we use the term 'analytical' we mean the 'numerically exact' procedure outlined above.

Apart from this solution of the dynamics, we can calculate the population of energy eigenstates numerically with TCSA. This is a benchmark for our numerical method as it is contrasted with a numerically exact calculation. We can compare Eq. (3.6) with Eq. (3.3) to express the overlap $g$ of a state which consists of only a single particle pair with momentum $p$:

$$|\langle p, -p|\Psi(t)\rangle|^2 \equiv |g_p(t)|^2 = n_p(t)\prod_{p' \neq p}(1 - n_{p'}(t)), \qquad (3.7)$$

where the product goes over the infinite set of quantised momenta in finite volume. It is straightforward to generalise Eq. (3.7) to express the overlap of any state with the pair structure of the free spectrum with the time-evolved state.

In practice, we truncate this product at some finite $p_{\max}$, since the goal is to match the analytic results with TCSA that operates with a truncation of its own. The one-mode cut-off of the analytic method and the many-body cut-off of TCSA cannot be brought to one-to-one correspondence with each other. However, overlaps are very sensitive to the number of states kept in each expansion, due to the constraint $\sum_n |g_n|^2 = 1$. Hence, our choice for the energy cutoff of TCSA for these figures is motivated by the goal to have the best possible match of the two approaches. Note that this is a single parameter for all the states.

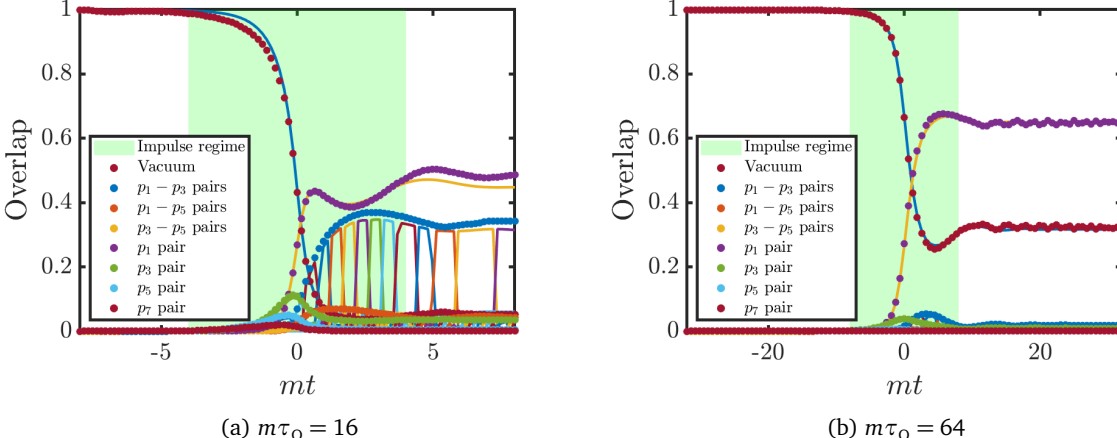

(a) $m\tau_Q = 16$          (b) $m\tau_Q = 64$

Figure 3.1: Overlaps of the evolving wave function with instantaneous eigenstates for two different ramps from the paramagnetic to the ferromagnetic phase with $m\tau_Q = 16$ and $m\tau_Q = 64$ for $mL = 50$ ($m = M_i$ in terms of the initial mass). The green region indicates the non-adiabatic regime. Solid lines are TCSA data for $N_{cut} = 25$ while dots are obtained from the numerical solution of the exact differential equations. Analytical results are plotted only for the few low-momentum states with the most substantial overlap. Lower indices in the legends refer to the quantum numbers of the modes present in the many-body eigenstate: $p_n = n\pi/L$. The composite structure of some lines is caused by level crossings experienced by multiparticle states.

The time evolution of the overlaps is presented in Fig. 3.1. Dots correspond to the solution of the differential equations for each mode and continuous lines denote TCSA data obtained by solving the many-body dynamics numerically. Fig. 3.1a depicts a curious behaviour of the second largest overlap in TCSA: the corresponding line seemingly consists of many different segments. This is a consequence of level crossings and the errors of numerical diagonalisation near these crossings. The state in question consists of two two-particle pairs and as the mass scale $M$ is ramped its energy increases steeper than that of high-momentum states with only a single pair, hence the level crossings. At each crossing the numerical diagonalisation cannot resolve precisely levels in the degenerate subspace, so the resulting overlap is not accurate. This accounts for the most prominent difference between the numerical and analytical results. Apart from that, the agreement is quite satisfactory.

The light green background corresponds to the naive impulse regime $t \in [-\tau_{KZ}, \ \tau_{KZ}]$. Of course this is only a crude estimate for the time when adiabaticity breaks down as Eq. (3.5) is strictly valid only as a scaling relation. Nevertheless, most of the change in each state population indeed happens within this coloured region. This statement is even more accentuated by Fig. 3.1b, that is, for a slower ramp. Comparing the two panels of Fig. 3.1 we observe that increasing the ramp time the probability of adiabaticity increases while the weight of the multiparticle states are suppressed. Note that although the two lowest available levels (the ground state and the first excited state) dominate the time-evolved state, the dynamics is far from being completely adiabatic that would mean no excitations at all. Hence, in accordance with the remarks concerning finite size effects in Sec. 2.1, we are within the regime of Kibble–Zurek scaling instead of being adiabatic.

We can also calculate the energy resolved version of the above figures, i.e. the instantaneous statistics of work, $P(W, t)$. We present this quantity in Fig. 3.2. The different ridges cor-

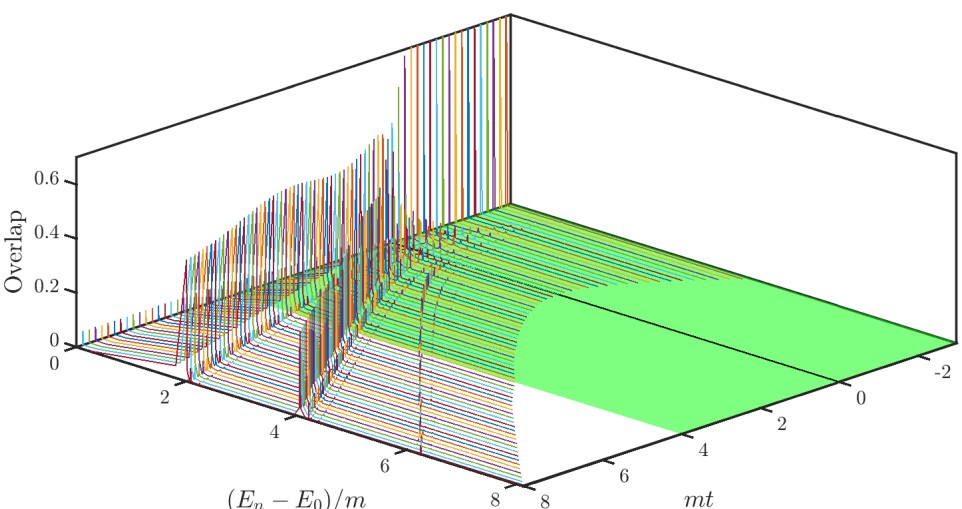

Figure 3.2: Instantaneous statistics of work $P(W, t)$ along a ramp with $m\tau_Q = 16$ from the paramagnetic to the ferromagnetic phase for $mL = 50$, obtained by TCSA with $N_{cut} = 45$. The height corresponds to the time-dependent overlap squares. The green region indicates the non-adiabatic regime.

respond to "bands" of 2-particle, 4-particle etc. states with energy thresholds $E = 2M, 4M, \ldots$. The ridges diverge linearly in time, displaying the linear dependence of the gap on the linearly tuned $M$ coupling. This figure illustrates the validity of the KZ arguments: low-energy bands dominate the excitations, and in each band, the modes with the lowest momenta (longest wavelengths) near the thresholds are the most prominent. This feature is similar to what was observed on the lattice in Ref. [39].

### 3.1.2 The ferromagnetic-paramagnetic (FP) direction

The ferromagnetic ground state is twofold degenerate in infinite volume. For the initial state we choose the state with maximal magnetisation corresponding to the infinite volume symmetry breaking state: $|\Psi_0\rangle = \frac{1}{\sqrt{2}}(|0\rangle_R + |0\rangle_{NS})$. As both sectors are present in the initial state, the time-evolved state also overlaps with both sectors. This provides yet another benchmark for our numerical approach and also a somewhat richer landscape of the overlap functions.

As one can see in Fig. 3.3, the dynamics are very similar to the PF case with the main difference coming from the fact that both sectors contribute. The different behaviour of the two vacua stems from the different available momentum modes in each sector: in the Ramond sector the momenta are larger in the lowest available modes and consequently they are less likely to be excited.

## 3.2 Ramps along the $E_8$ line

After investigating the free fermion line, we now turn to the behaviour of overlaps in the other integrable direction, i.e. for ramps along the $E_8$ axis defined by the protocol

$$h(t) = -2h_i t/\tau_Q \qquad (3.8)$$

for $t \in [-\tau_Q/2, \tau_Q/2]$. The scaling dimension of the perturbing operator $\sigma$ is $\Delta_\sigma = 1/8$, so critical exponent $\nu$ is different in this direction from the free fermion case: $\nu = 1/(2 - \Delta_\sigma) = 8/15$

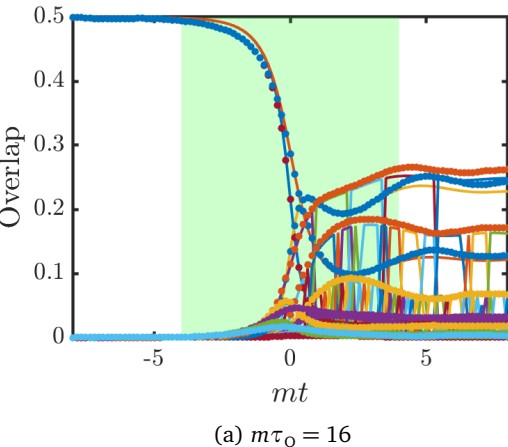

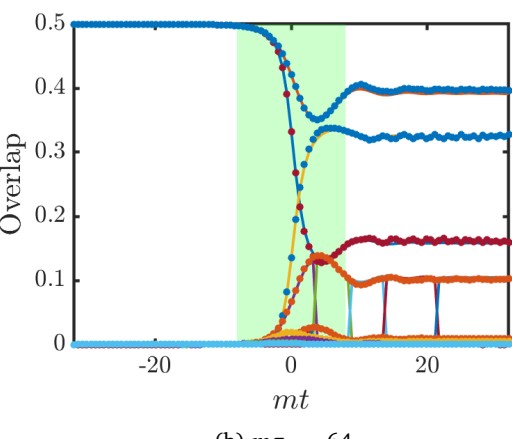

(a) $m\tau_Q = 16$                               (b) $m\tau_Q = 64$

Figure 3.3: Overlaps of the evolving wave function with instantaneous eigenstates for two different ramps from the ferromagnetic to the paramagnetic phase with $m\tau_Q = 16$ and $m\tau_Q = 64$ for $mL = 50$ ($m = -M_i$ in terms of the initial mass). The green region indicates the non-adiabatic regime. Solid lines are TCSA data for $N_{cut} = 31$ while dots are obtained from the numerical solution of the exact differential equations. Multiple pair states show several level crossings.

(cf. Eq. (2.14)). This implies that the Kibble–Zurek time (2.2) is given by

$$m_1\tau_{KZ} = \left(m_1\tau_Q\right)^{8/23} , \tag{3.9}$$

where, similarly to the free fermion case, the choice of the proportionality factor being 1 is just a convention.

    Let us first take an overview of the dynamics by looking at the time-dependent work statistics $P(W, t)$ shown in Fig. 3.4.

    Notice that in accordance with the Kibble–Zurek scenario, predominantly low-energy and low-momentum modes get excited in the course of the ramp. In the $E_8$ theory with multiple stable particles, the time evolved state has finite overlap not only with states consisting of pairs but also with states containing standing particles with zero momentum, including multiparticle states with a single such particle. We can observe that the energy distribution has peaks at some finite energy values, but low-momentum modes dominate for all branches (denoted by dashed lines of the same colour). This can be seen more clearly in Fig. 3.5 which presents $P(W)$ at the end of two ramps that differ in duration. Solid vertical lines indicate the energies of states consisting of standing particles only, i.e. combinations of particle masses.

    Let us remark that the perturbative calculations indicate that the KZ scaling applies to the overlap of each one-particle state and two-particle branch separately. That is a nontrivial statement since the spectrum of the $E_8$ field theory is a result of a bootstrap procedure relying heavily on delicate details of the interaction, however, these details are overlooked by a first order perturbative calculation. Although we expect that the summed contribution of one- and two-particle states to the energy density satisfies the KZ scaling (in line with the generic reasoning of Sec. 2.1), the much stronger statement of APT concerning the scaling behaviour of separate branches does not necessarily hold true. This is in fact what we observe in Fig. 3.5: as the average excess heat diminishes, the overlap of low-lying states increase instead of decreasing. However, as we are going to show below, both quench times are within the KZ scaling region and the scaling of the excess heat does satisfy Eq. (2.6). A remote analogy can be drawn with the form factor series expansion calculation of the central charge in integrable

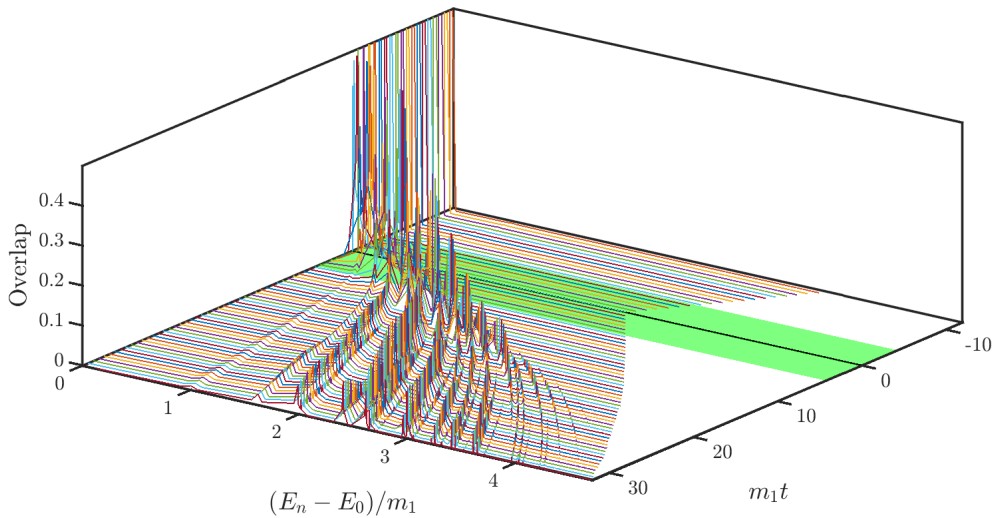

Figure 3.4: Instantaneous statistics of work $P(W, t)$ for a ramp along the $E_8$ axis with $m_1 \tau_Q = 64$, $m_1 L = 50$, obtained by TCSA with $N_{\text{cut}} = 45$. The height corresponds to the time-dependent overlap squares. The green region indicates the non-adiabatic regime. Notice the curvature of the "ridges" corresponding to the nonlinear $m_1 \propto h^{8/15}$ dependence of the mass gap on the distance from the critical point.

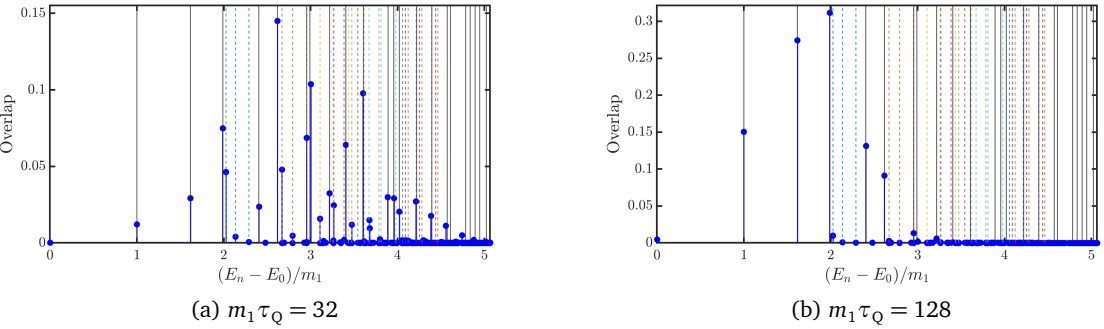

(a) $m_1 \tau_Q = 32$          (b) $m_1 \tau_Q = 128$

Figure 3.5: Statistics of work after the ramp $P(W, t = \tau_Q/2)$ along the $E_8$ direction with $m_1 L = 40$, $N_{\text{cut}} = 45$. States containing only zero-momentum particles are denoted by continuous lines, while dashed lines denote different moving multiparticle states.

perturbed conformal field theories, where the result of the sum over multiparticle states is fixed by the $c$-theorem, while the separate terms vary greatly due to the details of the interaction [99]. We note that in the current case the ambiguity arises from taking the $L \to \infty$ limit, since strictly speaking the adiabatic perturbation theory is sensible only if the ground state overlap remains close to 1, which is impossible for a finite density state in the thermodynamic limit. Previous calculations within the APT framework illustrate that this condition can be relaxed when calculating intensive quantities [19, 40], demanding a low-density time-evolved state instead of one with almost unity overlap with the instantaneous ground state. Although this approach successfully captures qualitative features of the KZ scaling, the above considerations indicate that one has to be careful as to what extent to draw conclusions from it.

### 3.3 Probability of adiabaticity

To study the Kibble–Zurek scaling using the TCSA, it is important to identify the time scale on which it is valid. For a finite volume method the time scale is limited from above by the onset of adiabaticity (cf. Eq. (2.9)) and also from below due to the natural time scale of the theory that is related to the mass gap before and after the ramp. A control quantity that can be used to fix the domain of $\tau_Q$ where the Kibble–Zurek scaling applies is the probability to be adiabatic after the ramp, $P(0, t_f)$. This overlap is exponentially suppressed with the volume, but its logarithm is proportional to the density of quasiparticles $n_{ex}$, such that $-\log(P(0))/L \propto n_{ex}$. Within the domain of validity for the Kibble–Zurek scaling the density scales according to Eq. (2.4), i.e. decays as a power law with $\tau_Q$. However, at the onset of adiabaticity it is exponentially suppressed [6, 13]. To explore the time scale mentioned above connected to volume parameters available for our calculation, we investigate the logarithm of the ground state overlap $P(0)$ after the ramp.

For ramps along the free fermion line there are two ways to evaluate $P(0)$. The first follows from the numerically exact solution of the problem in the scaling limit (see Appendix B). Second, we can use TCSA to calculate the ground state overlap. The onset of adiabaticity occurs at different quench times $\tau_Q$ depending on the volume parameter. Then the claim that for a given volume $L$ we can observe the KZ scaling – as opposed to adiabatic behaviour – can be supported by the observation that changing the volume does not alter the KZ scaling. Fig. 3.6a presents the comparison of the two methods with the slope of the KZ scaling as a guide to the eye. Apart from the very fast ramps, the two methods coincide with each other. We note that the onset of adiabaticity signalled by the strong deviation of different volume curves from each other and from the $\tau_Q^{-1/2}$ line is not an abrupt change but rather a smooth crossover. Nevertheless, we can identify that for $m\tau_Q \approx 5 \cdot 10^0 \dots 10^2$ the Kibble–Zurek scaling is satisfied to a good precision using the volume parameters available to the numerical method.

In the $E_8$ model we can only resort to the results of TCSA. Fig. 3.6b shows that the logarithm of the ground state overlap scales as the density of quasiparticles for large enough $\tau_Q$. Although the KZ scaling sets in later, i.e. for larger $\tau_Q$ than in the free fermion case, it is persistent up to the maximum ramp duration available to our numerical method. This is due to the fact that the exponent appearing in Eq. (2.9) is larger for the $E_8$ model and consequently the onset of adiabaticity occurs for a slower ramp in the same volume.

### 3.4 Ramps ending at the critical point

As detailed in Section 2.3, we expect the generic scaling arguments of APT for the Kibble–Zurek mechanism (Eqs. (2.24) and (2.25)) to be valid for ramps along both integrable lines of the model. A direct consequence of this claim is that the high-energy tail of the function $|K(\eta)|^2$ decays as $\eta^\beta$ with $\beta = -2z - 2/\nu$ (cf. Eq. (2.28)). This behaviour is important in view of the convergence properties of the integrals of the form (2.26).

To investigate the decay of high-energy overlaps with TCSA, we consider ramp protocols along the two integrable lines of the parameter space that end at the conformal point (ECP ramps). There are two reasons for this choice of protocol: first, TCSA uses the conformal basis and hence expected to be the most accurate at the critical point. Second, the dispersion relation is $E(k) = |k|$ in this case, so the high-energy tail of $P(W)$ decays with the same power law as $|\alpha(k)|^2$. Since $k$ and $\eta$ are related by a simple rescaling with the appropriate power of $\tau_Q$, the high energy tail of $P(W)$ should decay as $W^\beta$ at the critical point as far as the perturbative approach is correct, i.e. for slow enough ramps.

On the free fermion line we have $z = \nu = 1$, so $\beta = -2z - 2/\nu = -4$, while for an $E_8$ ramp $\nu = 8/15$ and the predicted exponent of the decay is $\beta = -23/4$. We remark that this can be contrasted with the high-energy tail of pair overlaps for sudden quenches. For quenches along

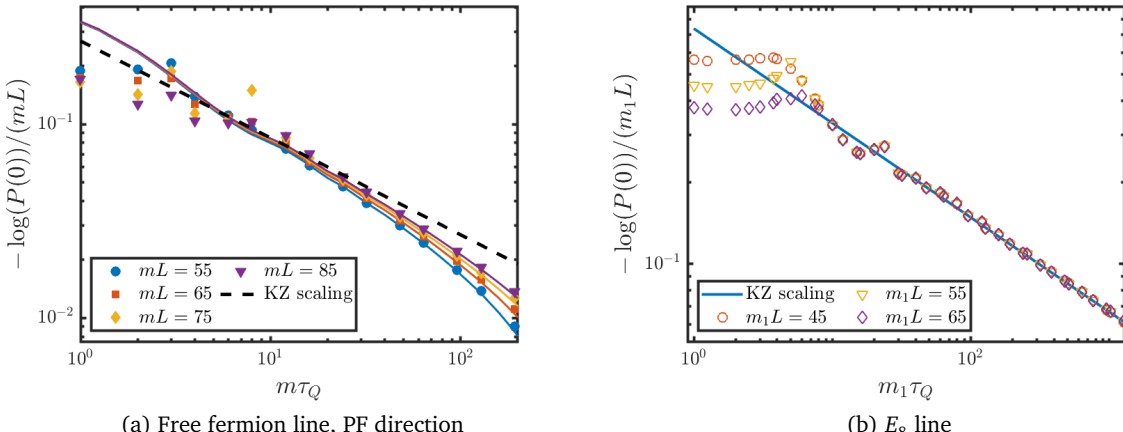

(a) Free fermion line, PF direction

(b) $E_8$ line

Figure 3.6: Logarithm of the probability of adiabaticity after a linear ramp along the two integrable lines of the Ising Field Theory. (a) Continuous lines and symbols of the same colour denote analytical and extrapolated TCSA data, respectively for various volume parameters. Black dashed line denotes the KZ scaling. At the onset of adiabaticity finite volume results deviate from the KZ slope and each other in a more pronounced manner. (b) Symbols stand for extrapolated TCSA data and the slope of the continuous line signals the KZ scaling exponent.

the free fermion line the exact solution yields $\beta = -2$ [81, 100, 101], while in the $E_8$ model the high energy tail of the perturbative expression decays with $\beta = -15/4$ [92], so $\beta = -2/\nu$ in both cases. The additional term of $-2z$ is the result of the adiabatic driving which suppresses the excitation of high energy modes.

In Fig. 3.7 we present the TCSA data and the slope of the straight line fitted to the logarithmic data. The two exponents are well separated and captured approximately correctly by the data. Let us note that the three highest-energy overlaps for each quench rate $\tau_Q$ do not follow the power-law decay, in fact, they are several orders of magnitude larger than the overlap of states with a slightly lower energy (cf. Fig. 3.7b). This an artefact of truncation: for any cut-off parameter the three overlaps corresponding to the largest available conformal cut-off level are anomalous in the above sense. However, for different cut-off parameters the outlying states have different energy, hence this is not a physical effect and the corresponding states are left out of the fit capturing the power-law decay.

We remark that Fig. 3.7a is analogous to Fig. 2c of Ref. [39] that reported a $W^{-8}$ decay. This is at odds with the prediction deduced from generic scaling arguments using APT and also with our TCSA results that favor the $\beta = -4$ exponent. Fig. 3.7 is in agreement with the numerous observations [7, 16, 26, 40] that adiabatic perturbation theory captures the correct Kibble–Zurek scaling in the free fermion theory and demonstrates that it applies also in the interacting $E_8$ integrable model. This is evidence that the arguments of APT can be generalised to this nontrivial theory which in turn implies that the Kibble–Zurek scaling can be observed there as well.

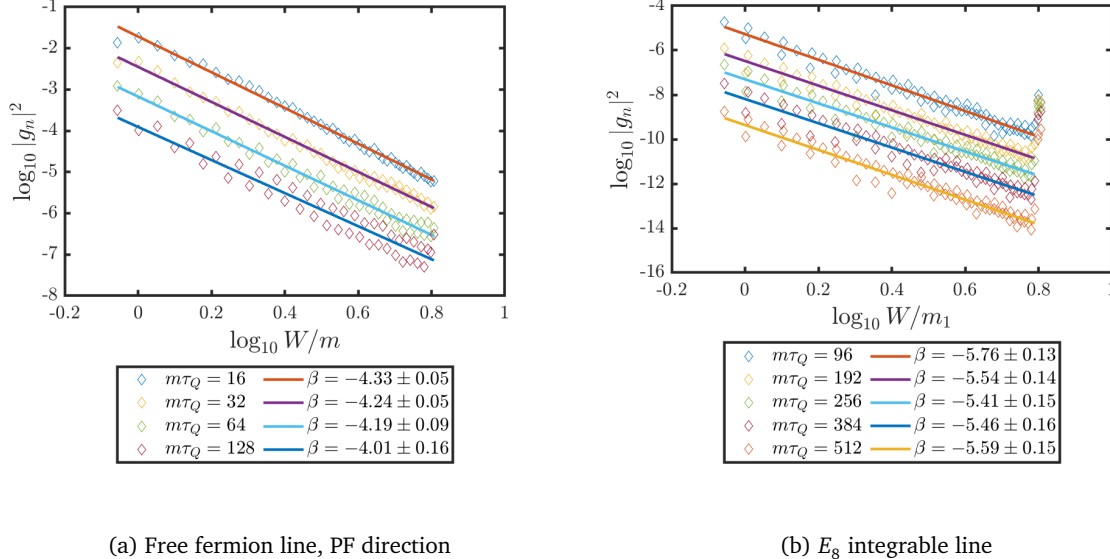

(a) Free fermion line, PF direction

(b) $E_8$ integrable line

Figure 3.7: High-energy overlaps for ramp protocols ending at the critical point with $mL = 50$, $N_{\text{cut}} = 51$. Data from different ramp rates are shifted vertically for better visibility. The slopes are linear fits of the logarithmic data and are close to the exponents predicted by APT: $\beta_{\text{FF}} = -4$ and $\beta_{E8} = -5.75$. Outlying highest-energy overlaps are omitted from the linear fit.

## 4 Dynamical scaling in the non-adiabatic regime

In this section we explore the dynamical scaling aspect of the Kibble–Zurek mechanism in the Ising Field Theory considering two one-point functions. We focus on the energy density and the magnetisation, both of which are important observables in the theory.

The energy density over the instantaneous vacuum or the excess heat density is defined as

$$w(t) = \frac{1}{L} \langle \Psi(t)|H(t) - E_0(t)|\Psi(t) \rangle, \tag{4.1}$$

where the Hamiltonian $H(t)$ has an explicit time dependence governed by the ramping protocol and $E_0(t)$ is the ground state of the instantaneous Hamiltonian $H(t)$. In accordance with Eq. (2.6), the excess heat for different ramp rates is expected to collapse to a single scaling function:

$$w(t/\tau_{\text{KZ}}) = \xi_{\text{KZ}}^{-d-\Delta_H} F_H(t/\tau_{\text{KZ}}) = \tau_{\text{KZ}}^{-d/z-1} F_H(t/\tau_{\text{KZ}}) = \tau_{\text{KZ}}^{-2} F_H(t/\tau_{\text{KZ}}), \tag{4.2}$$

where $d = 1$ is the spatial dimension, $\Delta_H = z$ is the scaling dimension of the energy and the second equation follows from $\tau_{\text{KZ}} = \xi_{\text{KZ}}^z$. For ramps along the free fermion line the energy density can be obtained from the solution of the exact differential equations using the mapping to free fermions, yielding essentially exact results.

The magnetisation operator $\sigma$ that corresponds to the order parameter has scaling dimension $\Delta_\sigma = 1/8$ hence is expected to satisfy the following scaling in the impulse regime ($z = 1$):

$$\langle \sigma(t/\tau_{\text{KZ}}) \rangle = \tau_{\text{KZ}}^{-1/8} F_\sigma(t/\tau_{\text{KZ}}). \tag{4.3}$$

In contrast to the energy density, the magnetisation is much harder to calculate even in free fermion case as it is a highly non-local operator in terms of the fermions.

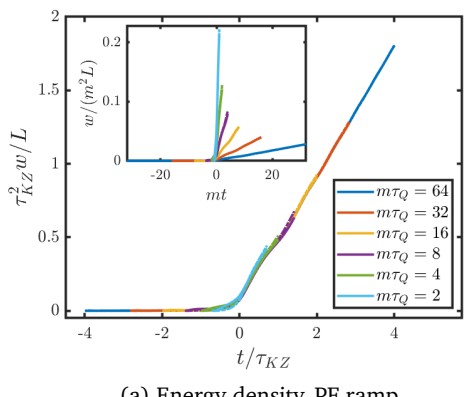
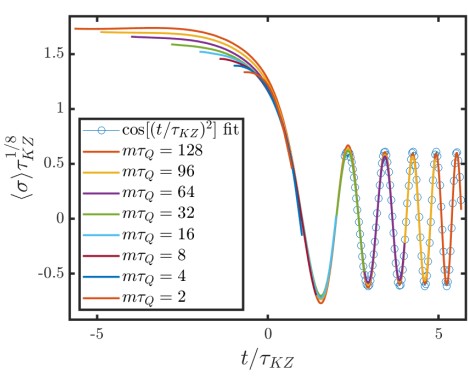

(a) Energy density, PF ramp          (b) Order parameter, FP ramp

Figure 4.1: Dynamical scaling of the energy density and the magnetisation for ramps along the free fermion line. Solid lines denote exact analytical solution while dot-dashed lines represent TCSA results for $mL = 50$ extrapolated in the cutoff. (a) Energy density along ramps of different speed in the paramagnetic-ferromagnetic direction. Inset illustrates the need for rescaling. (b) KZ scaling of the magnetisation $\sigma$ in the ferromagnetic-paramagnetic direction. The fitted function corresponding to the instantaneous one-particle oscillation is $f(t/\tau_{KZ}) = 0.612(2)\cos\left((t/\tau_{KZ})^2 + 0.830(3)\right)$. (Note that $(t/\tau_{KZ})^2 = m(t)t$.)

## 4.1 Free fermion line

We start with the free fermion line where exact analytical results are available. In Fig. 4.1a we observe the scaling behaviour (4.2) for several ramps from the paramagnetic to the ferromagnetic phase. Both the analytic calculations and the TCSA data, extrapolated in the cutoff, retain the scaling and the numerics agree almost perfectly with the exact results. The inset shows that the non-rescaled curves deviate substantially from each other.

As Fig. 4.1a shows, the collapse of the curves is perfect even well beyond the end of the non-adiabatic regime, in agreement with the observation and arguments of Ref. [33]. This can be understood in view of the eigenstate dynamics presented in Sec. 3. The relative population of energy eigenstates does not change substantially in the post-impulse regime and the increase in energy density then is merely due to the increasing gap $\Delta(t)$ as the coupling is ramped. The energy scale increases identically for all quench rates which in turn leads to the collapse of different curves. This argument can be formalised for the general setup of Sec. 2.1 as

$$w(t \gg \tau_{KZ}) \approx n_{ex}(t) \cdot \Delta(t) \propto \tau_{KZ}^{-d/z}\left(\frac{t}{\tau_Q}\right)^{a\nu z} \propto \tau_{KZ}^{-d/z}\left(\frac{t}{\tau_{KZ}}\right)^{a\nu z}\tau_{KZ}^{-1}, \qquad (4.4)$$

where $n_{ex}$ is the density of defects that is constant well beyond the impulse regime and scales as $\tau_{KZ}^{-d/z}$. The gap scales as $(t/\tau_Q)^{z\nu}$ and we used that $(\tau_{KZ}/\tau_Q)^{a\nu z} \propto \tau_{KZ}^{-1}$. The result shows that $w(t \gg \tau_{KZ})$ is a function of $t/\tau_{KZ}$. In the present case $a = \nu = z = 1$, which explains the linear behaviour seen in Fig. 4.1a.

The scaling behaviour of the magnetisation (4.3) is checked in Fig. 4.1b. The scaling is present most notably in terms of the frequency of the oscillations beyond the non-adiabatic window. Due to truncation errors of the TCSA method (see Appendix C), the predicted scaling is not reproduced perfectly in terms of the amplitudes and neither in the first half of the non-adiabatic regime. This is also the reason why the various curves do not collapse perfectly for times $t < -\tau_{KZ}$ where the scaling should also hold according to Eq. (2.7).

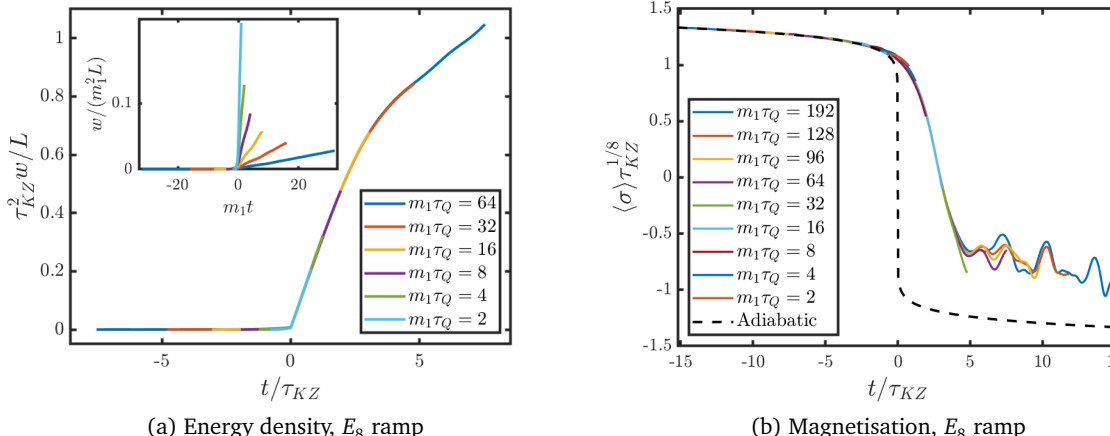

(a) Energy density, $E_8$ ramp

(b) Magnetisation, $E_8$ ramp

Figure 4.2: Dynamical scaling of the (a) energy density and (b) magnetisation in finite ramps across the critical point along the $E_8$ axis. The TCSA results obtained for $m_1 L = 50$ are extrapolated in the energy cutoff. The Kibble–Zurek scaling is present with $\tau_{\text{KZ}} \sim \tau_{\text{Q}}^{8/23}$. In panel (a) the inset shows the 'raw' curves without rescaling. In (b) the dashed black line shows the exact adiabatic value [102]: $\langle \sigma \rangle_{\text{ad}} = (-1.277578\ldots) \cdot \text{sgn}(h) |h|^{1/15}$.

The frequency of the late time oscillations is increasing with time. The oscillations can be fitted with the function $f(t) = A \cos[m(t) \cdot t + \phi]$ which demonstrates that the oscillations originate from one-particle states whose masses and thus the frequency increases in time with the gap. We remark that this is analogous to sudden quenches in the Ising Field Theory where the presence of one-particle oscillations is supported by analytical and numerical evidence [81, 84, 100]. The oscillations appear undamped well after the impulse regime $t/\tau_{\text{KZ}} \gg 1$. We remark that for sudden quenches the decay rate of the oscillations depends on the post-quench energy density [100, 101]. We expect the same to apply for ramps as well, but here the energy density is suppressed for slower ramps so the damping cannot be observed during a finite ramp. In contrast, the decay of oscillations in the dynamics of the order parameter after the ramp is observed in Ref. [39] in the spin chain.

## 4.2 Ramps along the $E_8$ axis

The dynamical scaling is well understood for the free fermion model on the lattice, and in the previous sections we demonstrated that they apply in the continuum scaling limit as well. The same aspect of the other integrable direction of the Ising Field Theory is yet unexplored. We now present how the simple scaling arguments of the KZM apply in a strongly interacting model. The dynamics in the $E_8$ model cannot be treated exactly due to the interactions but the numerical method of TCSA can be applied to simulate the time evolution. Truncation errors are expected to be less substantial since the $\sigma$ perturbation of the CFT is more relevant and exhibits faster convergence compared to the free fermion model (cf. Fig. 3.7). Hence using the conformal eigenstates as a basis of the Hilbert space is expected to be a better approximation.

As discussed above, the scaling is modified compared to the free fermion model due to the different exponent $\nu = 8/15$, so the Kibble–Zurek time scale $\tau_{\text{KZ}}$ depends on the ramp time $\tau_{\text{Q}}$ as $\tau_{\text{KZ}} = \tau_{\text{Q}}^{8/23}$. We demonstrate this scaling in the following for the dynamics of the energy density and the magnetisation.

Let us first discuss the scaling of the energy density presented in Fig. 4.2a. Similarly to the free fermion case, one observes an almost perfect collapse of the curves after crossing the

critical point, and the collapse is sustained beyond the impulse regime where now Eq. (4.4) predicts a $\sim (t/\tau_{\text{KZ}})^{8/15}$ behaviour.

Note that the above argument relies on the fact that the scaling properties of the energy density can be determined by considering it as the product of some defect density and a typical energy scale. For more complex quantities, such as the magnetisation for example, a similar argument does not apply, as Fig. 4.2b demonstrates. The curves deviate after the non-adiabatic regime but the collapse in the early adiabatic regime is perfect.

## 5   Cumulants of work

So far we have gained insight in the KZM by examining the instantaneous spectrum directly and demonstrated the relevance of the Kibble–Zurek time scale in dynamical scaling functions of local observables. In this section we aim to demonstrate that the Kibble–Zurek scaling is present in an even wider variety of quantities: the full statistics of the excess heat (or work) during the ramp is subject to scaling laws of the KZ type as well.

A particularly interesting result of the free fermion chain (already tested experimentally, cf. Ref. [48]) is that apart from the average density of defects and excess heat, their full counting statistics is also universal in the KZ sense: all higher cumulants of the respective distribution functions scale according to the Kibble–Zurek laws [36, 40]. The scaling exponents depend on the protocol in the sense that they are different for ramps ending at the critical point (ECP) and those crossing it (TCP). As Ref. [37] demonstrates, the universal scaling of cumulants can be observed in models apart from the transverse field Ising spin chain, hence it is natural to explore their behaviour in the Ising Field Theory.

The cumulants of excess work are defined via a generating function $\ln G(s)$:

$$G(s) = \langle \exp[s(H(t) - E_0(t))] \rangle \,, \tag{5.1}$$

where the expectation value is taken with respect to the time-evolved state. The cumulants $\kappa_i$ are the coefficients appearing in the expansion of the logarithm:

$$\ln G(s) = \sum_{i=1}^{\infty} \frac{s^i}{i!} \kappa_i \,. \tag{5.2}$$

The first three cumulants coincide with the mean, the second and the third central moments, respectively. Assuming that the generating functions satisfy a large deviation principle [40, 103], all of the cumulants are extensive $\propto L$. Consequently, we are going to focus on the $\kappa_i/L$ cumulant densities.

Elaborating on the framework of adiabatic perturbation theory presented in Sec. 2.3, we can argue that the scaling behaviour of the cumulants of the excess heat are not sensitive to the presence of interactions in the $E_8$ model and take a route analogous to Ref. [40] to obtain the KZ exponents. The core of the argument is the following: the Kibble–Zurek scaling within the context of APT stems from the rescaling of variables (2.23) which yields Eq. (2.26) from Eq. (2.22). The rescaling concerns the momentum variable that originates from the summation over pair states.

Now consider that cumulants can be expressed as a polynomial of the moments of the distribution:

$$\kappa_n = \mu_n + \sum_{\lambda \vdash n} \alpha_\lambda \prod_{i=1}^{k} \mu_{n_i} \,, \tag{5.3}$$

where $\lambda = \{n_1, n_2, \ldots, n_k\}$ is a partition of the integer index $n$ with $|\lambda| = k \geq 2$, and $\alpha_\lambda$ are integer coefficients. The moments are defined for the excess heat as

$$\mu_n = \langle [H - E_0]^n \rangle \,. \tag{5.4}$$

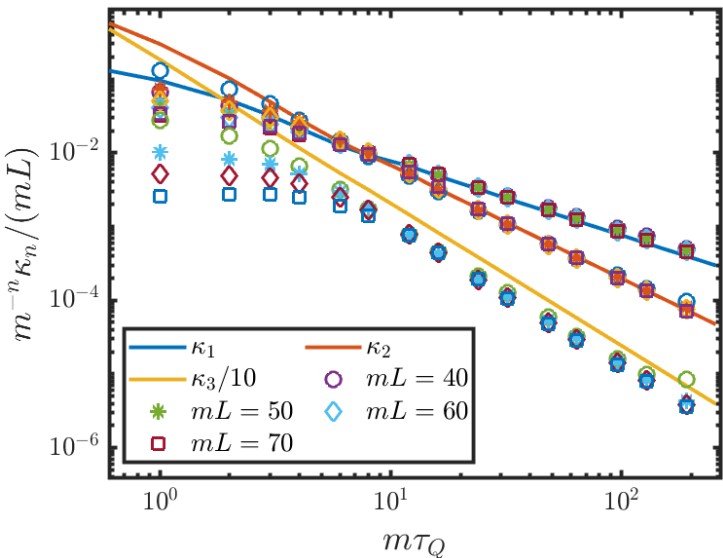

Figure 5.1: Cumulant densities for linear ramps on the free fermion line starting in the paramagnetic phase and ending at the QCP: a comparison between the numerically exact solution (solid lines) in the thermodynamic limit and cutoff-extrapolated TCSA data in different volumes (symbols). For both approaches $\kappa_3/L$ is plotted a decade lower for better visibility.

Let us note that the integration variable subject to rescaling in Eq. (2.23) originates from taking the expectation value. Consequently, in the limit $\tau_Q \to \infty$ terms consisting of powers of lower moments are suppressed compared to $\mu_n$, because they are the product of multiple integrals of the form (2.26). So the scaling behaviour of $\kappa_n$ equals that of $\mu_n$, which is defined with a single expectation value, hence its scaling behaviour is given by the calculation in Sec. 2.3. We remark that this line of thought is completely analogous to the arguments of Ref. [40]. According to the above reasoning, all cumulants of the work and quasiparticle distributions in the $E_8$ model should decay with the same power law as $\tau_Q \to \infty$.

To put the claims above to test, we follow the presentation of Ref. [40] and we discuss the two different scaling for the cumulants: first considering ramps that end at the critical point then examining ramps that navigate through the phase transition.

## 5.1 ECP protocol: ramps ending at the critical point

For ramps that end at the critical point one may apply the scaling form in (2.6) since the final time of such protocols corresponds to a fixed $t/\tau_{KZ} = 0$. The resulting naive scaling dimension of a work cumulant $\kappa_n$ is then easily obtained since it contains the product of $n$ Hamiltonians with dimension $\Delta_H = z = 1$. Consequently, we expect

$$\kappa_n/L \propto \tau_{KZ}^{-d/z-n} \propto \tau_Q^{-\frac{a\nu(d+nz)}{a\nu z+1}}, \tag{5.5}$$

where we used Eq. (2.2). However, the arguments of adiabatic perturbation theory [40] as outlined in Sec. 2.3 demonstrate that this naive scaling is true only if the corresponding quantity is not sensitive to the high-energy modes. However, using APT one can express the cumulants similarly to the defect density in Eq. (2.26). If the corresponding rescaled integral does not converge that means the contribution from high-energy modes cannot be discarded and the resulting scaling is quadratic with respect to the ramp velocity: $\tau_Q^{-2}$. The crossover

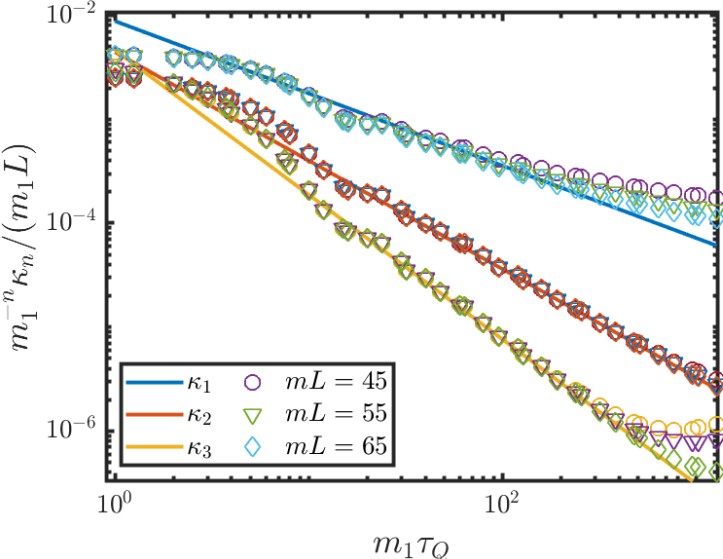

Figure 5.2: Cumulant densities for ECP ramps on the $E_8$ integrable line: cutoff-extrapolated TCSA data and the expected KZ scaling from dimension counting. The scaling exponents are 16/23, 24/23 and 32/23, respectively.

happens when $a\nu(d + nz)/(a\nu z + 1) = 2$; for smaller $n$ the KZ scaling applies while for larger $n$ quadratic scaling applies with logarithmic corrections at equality [24].

For the free fermion line $\nu = 1$ ($a = d = z = 1$) and the crossover cumulant index is $n = 3$. Fig. 5.1 justifies the above expectations for the three lowest cumulants by comparing the numerically exact solutions to TCSA results. TCSA is most precise for moderately slow quenches and the first two cumulants. There is notable deviation from the exact results in the case of the third cumulant although the scaling behaviour is intact. The deviation does not come as a surprise since the fact that the integral of adiabatic perturbation theory does not converge means that there is substantial contribution from all energy scales including those that fall victim to the truncation.

Fig. 5.1 also demonstrates that for very slow quenches finite size effects can spoil the agreement between exact results and TCSA. This is the result of the onset of adiabaticity (cf. Fig. 3.6a).

We expect identical scaling behaviour from the other integrable direction of the Ising Field Theory in terms of $\tau_{KZ}$ that translates to a different power-law dependence on $\tau_Q$. Indeed this is what we observe in Fig. 5.2. In this case there is no exact solution available hence solid lines denote the expected scaling law instead of the analytic result. The figure is indicative of the correct scaling although finite volume effects are more pronounced as the duration of the ramps is larger than earlier.

## 5.2 TCP protocol: ramps crossing the critical point

For slow enough ramps that cross the critical point and terminate at a given finite value of the coupling which lies far from the non-adiabatic regime where (2.6) applies, the excess work density scales identically to the defect density. This is due to the fact that the gap that defines the typical energy of the defects is the same for ramps with different $\tau_Q$ and the excess energy equals energy scale times defect density. It is demonstrated in Ref. [40] that higher cumulants of the excess work share a similar property: their scaling dimension coincides with that of the mean excess work, consequently all cumulants of the defect number and the excess

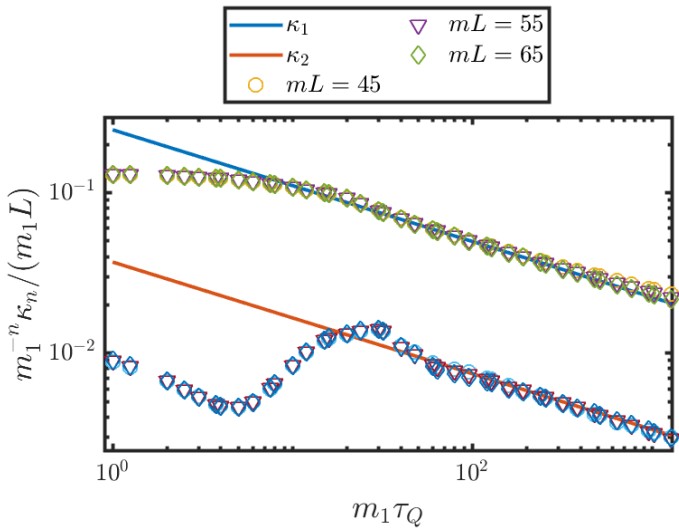

Figure 5.3: The first two cumulant densities for linear ramps crossing the QCP along the $E_8$ integrable line: the symbols represent cutoff-extrapolated TCSA data while the solid lines show the expected KZ scaling $\sim \tau_Q^{-8/23}$.

work scale with the same exponent. As we argued above, this claim is expected to be more general than free theories and in particular we claimed that it holds in the $E_8$ model.

Fig. 5.3 demonstrates the validity of this statement for the second cumulant. In line with the reasoning presented earlier (cf. Eq. (5.3) and below), the subleading terms are more prominent than in the case of the first cumulant (the excess heat) and KZ scaling is observable only for larger $\tau_Q$. Higher cumulants do not exhibit the same scaling within the quench time window available for TCSA calculations. Due to the increasing number of terms in the expressions with moments for the $n$th cumulant $\kappa_n$, we expect that the Kibble–Zurek scaling occurs for larger and larger $\tau_Q$, on time scales that are not amenable to effective numerical treatment as of now. Nevertheless, the behaviour of the second cumulant still serves as a non-trivial check of the assumptions that were used in Sec. 2.3 to apply APT to the $E_8$ model. As the argumentation did not rely explicitly on the details of the interactions in the $E_8$ theory, rather on the more general scaling behaviour of the gap (2.24) and the matrix element (2.25), we expect that a similar behaviour of the cumulants is observable in other interacting models exhibiting a phase transition.

# 6 Conclusions

In this paper we investigated the Kibble–Zurek scaling in the context of continuous quantum phase transitions in the Ising Field Theory. This model accommodates two types of universality in terms of the static critical exponent $\nu$ that corresponds to two integrable models for a specific choice of parameters in the space of couplings. One of them describes a free massive Majorana fermion and it exhibits a completely analogous KZ scaling to the transverse field Ising chain that can be mapped to free fermions. The second integrable direction corresponds to the famous $E_8$ model with its rich energy spectrum exhibiting eight stable particle states. Our main results concern the microscopic study of the KZ mechanism at the level of eigenstates, the time-dependent scaling of various observables, and the scaling of the cumulants of work.

In the free fermion direction, building on the lattice results, we expressed the nonequilibrium dynamics through the solution of a two-level problem and explored the Kibble–Zurek mechanism in terms of instantaneous eigenstates. We have shown that the adiabatic-impulse-adiabatic scenario is qualitatively correct at the most fundamental level of quantum state dynamics. That is, we can identify a non-adiabatic "impulse" regime where the most substantial change in the population of eigenstates happens, preceded and followed by a regime of adiabatic dynamics where these populations are approximately constant. We demonstrated that the relative length of the impulse regime compared to the duration of the ramp decreases as the time parameter of the ramp $\tau_Q$ increases, following the scaling forms dictated by the Kibble–Zurek mechanism. Although this simple picture has been investigated in earlier works, capturing it at the fundamental level of quantum states in a quantum field theory is still noteworthy.

We established parallelisms between the lattice and continuum dynamics for an extended set of scaling phenomena from the dynamical scaling of local observables to the universal behaviour of higher cumulants of the work. These analogies do not come as a surprise but their analysis in a field theoretical context is a novel result. Apart from generalising recently understood phenomena on the lattice to the continuum, these observations serve as a benchmark for our numerical method, the Truncated Conformal Space Approach. Comparing with analytical solutions available in the free fermion theory, we illustrated the capacity of this method to capture the intricate quantum dynamics behind the Kibble–Zurek scaling near quantum critical points. In spite of operating in finite volume, it is capable of demonstrating the presence of scaling laws within a wide interval of the time parameter $\tau_Q$ without substantial finite size effects. This is of paramount importance in the demonstration that the KZ scaling is not limited to the noninteracting dynamics within the Ising Field Theory.

One of the essential results of our work is that the Kibble–Zurek mechanism is able to account for the universal scaling of a strongly interacting theory, the $E_8$ model, near its quantum critical point. In order to have a solid case for this observation, we elaborated on the framework of adiabatic perturbation theory and applied its basic concepts to the $E_8$ model. While a refined version of the originally suggested adiabatic-impulse-adiabatic scenario predicts universal dynamical scaling of local observables in the non-adiabatic regime (which we also verified using TCSA, see Sec. 4), employing APT to address the nonequilibrium dynamics provides perturbative arguments also for the universal scaling of the full counting statistics of the excess heat and the number of quasiparticles. This reasoning has been used recently to explain the universal scaling of work cumulants in a free model [40]. In this work we have taken the next step and discussed its implications for the interacting $E_8$ field theory. We argued that the interactions do not alter the universal scaling of cumulants and demonstrated this in Sec. 5 for the first cumulants both for end-critical and trans-critical ramp protocols. We remark that our argument is in fact quite general and mostly relies on the small density induced by the nonequilibrium protocol. Since the KZ scaling predicts that the dynamics is close to adiabatic as $\tau_Q \to \infty$, this is a sensible assumption. Consequently, the result is expected to hold generally, i.e. all cumulants of the excess work should scale with the scaling exponents predicted by adiabatic perturbation theory irrespective of the interactions in the model.

This claim can be put to test in various experimental settings, e.g. using Rydberg atoms [45] to explore the Ising universality class, or in cold-atomic gases realising an interacting field theory [104]. Analogously to the universality established in the full counting statistics of kinks [48], we expect that similar signatures can be observed for the cumulants of the excess heat in an experiment that realises a genuinely interacting quantum system.

We note that there are several possible future directions. It is particularly interesting to test the scaling behaviour of "fast but smooth" ramps versus sudden quenches in the coupling space of field theoretical models [105–108]. The presence of universal scaling at fast quench rates

is remarkable though to implement an infinitely smooth ramp in an interacting theory that is not amenable to exact analytic treatment is not trivial. Another fruitful direction to take is the exploration of nonintegrable regimes within the Ising Field Theory and examine the interplay between the physics related to integrability breaking and the Kibble–Zurek scenario. Our findings suggest that the latter is in fact quite general but its validity in a generic non-integrable scenario remains to be tested.

## Acknowledgments

The authors are indebted to Gábor Takács for insightful discussions and comments on the manuscript. They also thank Anatoli Polkovnikov for useful correspondence.

**Funding information** The authors were partially supported by the National Research Development and Innovation Office of Hungary under the research grants OTKA No. SNN118028 and K-16 No. 119204, and also by the BME-Nanotechnology FIKP grant of ITM (BME FIKP-NAT). M.K. acknowledges support by a "Bolyai János" grant of the HAS, and by the "Bolyai+" grant of the ÚNKP-19-4 New National Excellence Program of the Ministry for Innovation and Technology.

## A  Application of the adiabatic perturbation theory to the $E_8$ model

To use the framework of adiabatic perturbation theory in the $E_8$ model we assume that the time-evolved state can be expressed as

$$|\Psi(t)\rangle = \sum_n \alpha_n(t) \exp\{-\imath \Theta_n(t)\} |n(t)\rangle \,, \tag{A.1}$$

with the dynamical phase factor $\Theta_n(t) = \int_{t_i}^t E_n(t') \, dt'$. We also assume that there is no Berry phase and thus to leading order in the small parameter $\dot{\lambda}$ the $\alpha_n$ coefficients take the form

$$\alpha_n(\lambda) \approx \int_{\lambda_i}^{\lambda} d\lambda' \langle n(\lambda') | \partial_{\lambda'} | 0(\lambda') \rangle \exp\{\imath (\Theta_n(\lambda') - \Theta_0(\lambda'))\}. \tag{A.2}$$

Higher derivatives as well as higher order terms in $\dot{\lambda}$ are neglected from now on.

The $\alpha_n$ coefficients can be used to formally express quantities that have known matrix elements on the instantaneous basis of the Hamiltonian:

$$\langle \mathcal{O}(t) \rangle = \sum_{m,n} \alpha_m^*(\lambda(t)) \alpha_n(\lambda(t)) \mathcal{O}_{mn} \,. \tag{A.3}$$

In what follows, we present the evaluation of this sum - approximately, under conditions of low energy density discussed in the main text - for the case of $\mathcal{O}(t) = H(t) - E_0(t)$ in the $E_8$ model. To generalise this calculation to the defect density or to higher moments of the statistics of work function is straightforward. The work density (or excess heat density) after the ramp reads

$$w(\lambda_f) = \frac{1}{L} \sum_n (E_n(\lambda_f) - E_0(\lambda_f)) |\alpha_n(\lambda_f)|^2 \,. \tag{A.4}$$

The spectrum of the model consists of 8 particle species $A_a, a = 1, \dots, 8$ with masses $m_a$. The energy and momentum eigenstates are the asymptotic states of the model labelled by a set of

relativistic rapidities $\{\vartheta_1, \vartheta_2, \ldots \vartheta_N\}$ and particle species indices $\{a_1, a_2, \ldots a_N\}$:

$$|n\rangle = |\vartheta_1, \vartheta_2, \ldots \vartheta_N\rangle_{a_1, a_2, \ldots a_N}, \tag{A.5}$$

with energy $E_n = \sum_{i=1}^N m_{a_i} \cosh(\vartheta_i)$ and momentum $p_n = \sum_{i=1}^N m_{a_i} \sinh(\vartheta_i)$. The summation in Eq. (A.4) in principle goes over the infinite set of asymptotic states. As discussed in the main text, for low enough density we can approximate the sum in Eq. (A.4) with the contribution of one- and two-particle states, analogously to the calculation in the sine–Gordon model in Ref. [17].

## A.1 One-particle states

Contribution of the one-particle states can be expressed as

$$w_{1\mathrm{p}} = \lim_{L \to \infty} \frac{1}{L} \sum_{a=1}^8 m_a |\alpha_a(\lambda_{\mathrm{f}})|^2, \tag{A.6}$$

where $m_a$ is the mass of the particle species $a$ and the summation runs over the eight species. We can write the coefficient $\alpha_a$ as

$$\alpha_a(\lambda_{\mathrm{f}}) = \int_{\lambda_{\mathrm{i}}}^{\lambda_{\mathrm{f}}} \mathrm{d}\lambda \, \langle \{0\}_a(\lambda)| \, \partial_\lambda \, |0(\lambda)\rangle \exp\left\{ \iota \tau_{\mathrm{Q}} \int_{\lambda_{\mathrm{i}}}^\lambda \mathrm{d}\lambda' m_a(\lambda') \right\}, \tag{A.7}$$

where $\langle \{0\}_a(\lambda)|$ denotes the asymptotic state with a single zero-momentum particle. The matrix elements and masses depend on $\lambda$ through the Hamiltonian that defines the spectrum. The matrix element can be evaluated as

$$\langle \{0\}_a(\lambda)| \, \partial_\lambda \, |0(\lambda)\rangle = -\frac{\langle \{0\}_a(\lambda)| \, V \, |0(\lambda)\rangle}{m_a(\lambda)}. \tag{A.8}$$

For an $E_8$ ramp that conserves momentum, $V$ is the integral of the local magnetisation operator $\sigma(x)$: $V = \int_0^L \sigma(x)\mathrm{d}x$. Utilising this we further expand

$$\langle \{0\}_a(\lambda)| \, \partial_\lambda \, |0(\lambda)\rangle = -\frac{L F_a^{\sigma*}(\lambda)}{m_a(\lambda)\sqrt{m_a(\lambda)L}}, \tag{A.9}$$

where the square root in the denominator emerges from the finite volume matrix element [109] and $F_a^\sigma$ is the (infinite volume) one-particle form factor of the magnetisation operator. It only depends on the coupling $\lambda$ through its proportionality to the vacuum expectation value of $\sigma$. The particle masses scale as the gap: $m_a(\lambda) = C_a|\lambda|^{z\nu}$, where $C_a$ are some constants. This allows us to write

$$|\alpha_a(\lambda_{\mathrm{f}})|^2 = L \left| \int_{\lambda_{\mathrm{i}}}^{\lambda_{\mathrm{f}}} \mathrm{d}\lambda \, \frac{\tilde{F}_a^{\sigma*}\lambda^{2\nu-1}}{C_a^{3/2}|\lambda|^{3/2z\nu}} \exp\left\{ \iota \tau_{\mathrm{Q}} \int_{\lambda_{\mathrm{i}}}^\lambda \mathrm{d}\lambda' C_a|\lambda'|^{z\nu} \right\} \right|^2. \tag{A.10}$$

We can perform the integral in the exponent that leads to a $\tau_{\mathrm{Q}}|\lambda|^{1+z\nu}$ dependence there. To get rid of the large $\tau_{\mathrm{Q}}$ factor in the denominator, we introduce the rescaled coupling $\zeta$ with

$$\zeta = \lambda \tau_{\mathrm{Q}}^{\frac{1}{1+z\nu}}. \tag{A.11}$$

The change of variables yields

$$|\alpha_a(\lambda_{\mathrm{f}})|^2 = L \tau_{\mathrm{Q}}^{-\frac{\nu(4-3z)}{1+z\nu}} \left| \int_{\zeta_{\mathrm{i}}}^{\zeta_{\mathrm{f}}} \tilde{C}_a \mathrm{sgn}(\zeta)|\zeta|^{2\nu-1-3/2z\nu} \exp\left\{ \iota C_a'|\zeta|^{1+z\nu} \right\} \right|^2, \tag{A.12}$$

where $\tilde{C}_a$ and $C'_a$ are constants that depend on $C_a$, the one-particle form factors and the critical exponents. We note the integral is convergent for large $\zeta$ due to the strongly oscillating phase factor and also for $\zeta \to 0$ since $2\nu - 1 - 3/2z\,\nu = -11/15$ in the $E_8$ model. Substituting $z = 1$ in the exponent of $\tau_Q$ leads to the correct KZ exponent of a relativistic model, $\nu/(1+\nu)$.

## A.2 Two-particle states

The contribution of a two-particle state with species $a$ and $b$ is going to be denoted $w_{ab}$ and reads

$$w_{ab}(\lambda_\mathrm{f}) = \frac{1}{L}\sum_\vartheta (m_a \cosh\vartheta + m_b \cosh\vartheta_{ab})|\alpha_\vartheta(\lambda_\mathrm{f})|^2\,, \tag{A.13}$$

where $\vartheta_{ab}$ is a function of $\vartheta$ determined by the constraint that the state has zero overall momentum. The summation goes over the rapidities that are quantised in finite volume $L$ by the Bethe–Yang equations:

$$Q_i = m_{a_i} L \sinh\vartheta_i + \sum_{j\neq i}^N \delta_{a_i a_j}(\vartheta_i - \vartheta_j) = 2\pi I_i\,, \tag{A.14}$$

where $I_i$ are integers numbers and

$$\delta_{ab} = -\imath \log S_{ab} \tag{A.15}$$

is the scattering phase shift of particles of type $a$ and $b$. For a two-particle state Eq. (A.14) amounts to two equations of which only one is independent due to the zero-momentum constraint. It reads

$$\tilde{Q}(\vartheta) = m_a L \sinh\vartheta + \delta_{ab}(\vartheta - \vartheta_{ab}) = 2\pi I\,, \qquad I \in \mathbb{Z}\,. \tag{A.16}$$

In the thermodynamic limit $L \to \infty$ the summation is converted to an integral with the integral measure $\frac{\mathrm{d}\vartheta}{2\pi}\tilde{\rho}(\vartheta)$, where $\tilde{\rho}(\vartheta)$ is the density of zero-momentum states defined by

$$\tilde{\rho}(\vartheta) = \frac{\partial \tilde{Q}(\vartheta)}{\partial\vartheta} = m_a L \cosh\vartheta + \left(1 + \frac{m_a \cosh\vartheta}{m_b \cosh\vartheta_{ab}}\right)\Phi_{ab}(\vartheta - \vartheta_{ab})\,, \tag{A.17}$$

where $\Phi(\vartheta)$ is the derivative of the phase shift function. The resulting integral is

$$\frac{1}{L}\int_{-\infty}^\infty \frac{\mathrm{d}\vartheta}{2\pi}\tilde{\rho}(\vartheta)|\alpha_\vartheta(\lambda_\mathrm{f})|^2\,. \tag{A.18}$$

The $\alpha_\vartheta(\lambda_\mathrm{f})$ term can be expressed as (cf. Eq. (A.2)

$$\alpha_\vartheta(\lambda_\mathrm{f}) = \int_{\lambda_\mathrm{i}}^{\lambda_\mathrm{f}} \mathrm{d}\lambda\, \langle\{\vartheta,\vartheta_{ab}\}_{ab}(\lambda)|\,\partial_\lambda\,|0(\lambda)\rangle \exp\left\{\imath\tau_Q \int_{\lambda_\mathrm{i}}^\lambda \mathrm{d}\lambda'\left[m_a(\lambda')\cosh\vartheta + m_b(\lambda')\cosh\vartheta_{ab}\right]\right\}\,. \tag{A.19}$$

Analogously to the one-particle case we can evaluate the matrix element in the $E_8$ field theory as

$$-\frac{{}_L\langle\{\vartheta,\vartheta_{ab}\}_{ab}(\lambda)|\,\sigma(0)\,|0(\lambda)\rangle_L}{E_n(\lambda) - E_0(\lambda)} = -\frac{LF_{ab}^{\sigma*}(\vartheta,\vartheta_{ab})}{(E_n(\lambda) - E_0(\lambda))\sqrt{\rho_{ab}(\vartheta,\vartheta_{ab})}}\,, \tag{A.20}$$

where $F_{ab}^\sigma(\vartheta_1,\vartheta_2)$ is the two-particle form factor of operator $\sigma$ in the $E_8$ field theory and the density factor is the Jacobian of the two-particle Bethe–Yang equations (A.14) arising from the normalisation of the finite-volume matrix element [109]. It can be expressed as

$$\rho_{ab}(\vartheta_1,\vartheta_2) = m_a L \cosh\vartheta_1 m_b L \cosh\vartheta_2 + (m_a L \cosh\vartheta_1 + m_b L \cosh\vartheta_2)\Phi_{ab}(\vartheta_1 - \vartheta_2)\,. \tag{A.21}$$

Observing Eqs. (A.17) and (A.21) one finds that the details of the interaction enter via the derivative of the phase shift function but crucially, they are of order $1/L$ compared to the free field theory part. So leading order in $L$ we find that

$$w_{ab}(\lambda_{\mathrm{f}}) = \int_{-\infty}^{\infty} \frac{\mathrm{d}\vartheta}{2\pi} (m_a(\lambda_{\mathrm{f}})\cosh\vartheta + m_b(\lambda_{\mathrm{f}})\cosh\vartheta_{ab}) m_a(\lambda_{\mathrm{f}})\cosh\vartheta \times$$
$$\times \left| \int_{\lambda_{\mathrm{i}}}^{\lambda_{\mathrm{f}}} \mathrm{d}\lambda \frac{F_{ab}^{\sigma*}(\vartheta,\vartheta_{ab})}{(m_a(\lambda)\cosh\vartheta + m_b(\lambda)\cosh\vartheta_{ab})\sqrt{m_a(\lambda)m_b(\lambda)\cosh\vartheta\cosh\vartheta_{ab}}} \times$$
$$(A.22)$$
$$\times \exp\left( \iota\tau_{\mathrm{Q}} \int_{\lambda_{\mathrm{i}}}^{\lambda} \mathrm{d}\lambda' \big(m_a(\lambda')\cosh\vartheta + m_b(\lambda')\cosh\vartheta_{ab}\big) \right) \Bigg|^2 + \mathcal{O}(1/L).$$

A change of variables in the outer integral to the one-particle momentum $p = m_a \sinh\vartheta$ we obtain

$$w_{ab} = \int_{-\infty}^{\infty} \frac{\mathrm{d}p}{2\pi} E_p(\lambda_{\mathrm{f}}) \left| \int \mathrm{d}\lambda\, G(\vartheta) \exp\left( \iota\tau_{\mathrm{Q}} \int \mathrm{d}\lambda' E_{\vartheta}(\lambda') \right) \right|^2. \qquad (A.23)$$

Now we can introduce the momentum $p$ in the inner integral as well by noting that the energy can be expressed as a function of momentum via the relativistic dispersion and that the relativistic rapidity also $\vartheta = \mathrm{arcsinh}(p/m)$. Since $m \propto |\lambda|^{z\nu}$ with $z = 1$ any expression that is a function of $\vartheta$ can be expressed as a function of $p/|\lambda|^{\nu}$. Having this in mind, the result is analogous to the free case so all the machinery developed there can be used. The key assumptions from this point regard the scaling properties of the energy gap and the matrix element $G(\vartheta)$ in this brief notation:

$$E_p(\lambda) = |\lambda|^{z\nu} F(p/|\lambda|^{\nu}) \qquad (A.24)$$
$$G(\vartheta) = \lambda^{-1} G(p/|\lambda|^{\nu}). \qquad (A.25)$$

These equations are trivially satisfied with the proper asymptotics for $F(x) \propto x^z$. For $G(x)$ one can verify using that in the $E_8$ model we have

$$\lim_{L\to\infty} \langle \{\vartheta,\vartheta_{ab}\}(\lambda)|\, \partial_{\lambda} |0(\lambda)\rangle_L = \frac{\langle\sigma\rangle F_{ab}^{\sigma*}(\vartheta,\vartheta_{ab})}{\sqrt{m_a\cosh\vartheta\, m_b\cosh\vartheta_{ab}}(m_a\cosh\vartheta + m_b\cosh\vartheta_{ab})}$$
$$= \lambda^{1/15-8/15-8/15} G(\vartheta) = \lambda^{-1} G(\vartheta), \qquad (A.26)$$

where we neglected the $\mathcal{O}(1/L)$ term from the finite volume normalisation and used $\langle\sigma\rangle \propto \lambda^{1/15}$, $m \propto \lambda^{8/15}$. $F_{ab}(\vartheta,\vartheta_{ab})$ is the two-particle form factor of the $E_8$ theory that does not depend on the coupling. They satisfy the asymptotic bound [99]:

$$\lim_{|\vartheta_i|\to\infty} F^{\sigma}(\vartheta_1,\vartheta_2\ldots,\vartheta_n) \leq \exp(\Delta_{\sigma}|\vartheta_i|/2). \qquad (A.27)$$

Since the matrix elements considered here are of zero-momentum states, $\vartheta \to \infty$ means $\vartheta_{ab} \to -\infty$ and $F_{ab}^{\sigma}(\vartheta,\vartheta_{ab}) \leq \exp(\Delta_{\sigma}\vartheta)$ as the form factors depend on the rapidity difference. Dividing by the factor $\exp(2\vartheta)$ in the denominator yields the correct asymptotics $G(x) \propto x^{\Delta-2} = x^{-1/\nu}$ as an upper bound due to Eq. (A.27). We remark that the scaling forms (A.24) hold true for any value of the coupling $\lambda$ in the field theory, in contrast to the lattice where they are valid only in the vicinity of the critical point. From this perspective Eq. (A.24) follows from the definition of the field theory as a low-energy effective description of the lattice model near its critical point.

As a consequence, one can introduce new variables in place of $\lambda$ and $p$ such that the explicit $\tau_Q$ dependence disappears from the integrand. This is achieved by the following rescaling:

$$\eta = p\tau_Q^{\frac{\nu}{1+z\nu}}, \qquad \zeta = \lambda\tau_Q^{\frac{1}{1+z\nu}}. \tag{A.28}$$

The result for the energy density is

$$w_{ab} = \tau_Q^{-\frac{\nu}{1+z\nu}} \int_{-\infty}^{\infty} \frac{\mathrm{d}\eta}{2\pi} E_{p=\eta\tau_Q^{-\frac{\nu}{1+z\nu}}}(\lambda_{\mathrm{f}}) |\alpha(\eta)|^2. \tag{A.29}$$

In terms of scaling there are two options: first, let $|\lambda_{\mathrm{f}}| \neq 0$ hence $\zeta_{\mathrm{f}} \to \infty$ in the KZ scaling limit $\tau_Q \to \infty$. Then the energy gap at $p \to 0$ is a constant and $E_{p=0}(\lambda_{\mathrm{f}})$ can be brought in front of the integral. If it converges, Eq. (A.29) completely accounts for the KZ scaling. Second, if $|\lambda_{\mathrm{f}}| = 0$, the energy gap is $E_p \propto p^z$ and an additional factor of $\tau_Q^{-\frac{\nu}{1+z\nu}}$ appears in front of the integral. Note that this is the scaling of $\kappa_1$ on Fig. 5.2. The high-energy tail of the integrand is modified due to the extra term of $\eta^z$ from the energy gap. This leads to a convergence criterion such that once again the crossover to quadratic scaling happens when the exponent of $\tau_Q$ in front of the integral is less then $-2$. It is easy to generalise this argument to the $n$th moment of the statistics of work which amounts to substituting $E_p^n$ instead of $E_p$ to Eq. (A.29). As argued in the main text, this is the leading term in the $n$th cumulant of the distribution as well, that concludes the perturbative reasoning behind the results of Sec. 5.

# B Ramp dynamics in the free fermion field theory

The non-equilibrium dynamics of the transverse field Ising chain is thoroughly studied in the literature. Due to the factorisation of the dynamics to independent fermionic modes solving the time evolution amounts to the treatment of a two-level problem parametrised by the momentum $k$. This two-level problem can be mapped to the famous Landau–Zener transition with momentum-dependent crossing time. Its exact solution is known and yields a particularly simple expression for the excitation probability of low-momentum modes $p_k$ (or $|\alpha(k)|^2$ with the notation of adiabatic perturbation theory, cf. Sec. 2.3) in the limit $\tau_Q \to \infty$. Then the KZ scaling of various quantities follows [8, 13] and extends to the full counting statistics of defects [36] and excess heat [40]. For a finite Landau–Zener problem one can express the solution in terms of Weber functions [26, 33] or for a generic nonlinear ramp profile as the solution of a differential equation [54, 103].

To generalise the analytical solution on the chain to the free field theory we performed the scaling limit on the expressions of Ref. [54]. We remark that in the works cited above there are several parallel formulations of this problem on the chain each with a slightly different focus. Our choice to use this specific one in the continuum limit is arbitrary but the result is the same for all frameworks. We use the following notation: $c_k^{(\dagger)}$ denotes the Fourier transformed fermionic (creation)-annihilation operators obtained by the Jordan–Wigner transformation. In each mode $k$, $\eta_k^{(\dagger)}$ are the quasiparticle ladder operators and we use $\eta_{k,\mathrm{i}}^{(\dagger)}$ to refer to the operators that diagonalise the Hamiltonian initially before the ramp procedure. The operators $c$ and $\eta$ are related via the Bogoliubov transformation

$$\eta_k = U_k c_k - \iota V_k c_{-k}^{\dagger}, \tag{B.1}$$

where the coefficients are $U_k = \cos\theta_k/2$ and $V_k = \sin\theta_k/2$ with

$$\exp(\iota\theta_k) = \frac{g - \exp(\iota k)}{\sqrt{1 + g^2 - 2g\cos k}}. \tag{B.2}$$

From a dynamical perspective $U$ and $V$ relate the adiabatic (instantaneous) free fermions and quasiparticles, hence we are going to refer to them as adiabatic coefficients. The dynamics can be solved in the Heisenberg picture using the Ansatz

$$c_k(t) = u_k(t)\eta_{k,\mathrm{i}} + \iota v^*_{-k}(t)\eta^\dagger_{k,\mathrm{i}}. \tag{B.3}$$

The Heisenberg equation of motion yields a coupled first order differential equation system for the time-dependent Bogoliubov coefficients that can be decoupled as [54]:

$$\frac{\partial^2}{\partial t^2}y_k(t) + \left(A_k(t)^2 + B_k^2 \pm \iota \frac{\partial}{\partial t}A_k(t)\right)y_k(t) = 0, \tag{B.4}$$

where the upper and lower signs correspond to $y_k(t) = u_k(t)$ and $y_k(t) = v^*_{-k}(t)$ respectively, and $A_k(t) = 2J(g(t) - \cos k)$ and $B_k = 2J \sin k$. To connect with the expression for the time-evolved $k$ mode in the main text,

$$|\Psi(t)\rangle_k = a_k(t)|0\rangle_{k,t} + b_k(t)|1\rangle_{k,t}, \tag{B.5}$$

we have to express $a_k(t)$ and $b_k(t)$ with the time-dependent Bogoliubov coefficients. To do so, first one has to perform a Bogoliubov transformation that relates the quasiparticle operators $\eta_{k,\mathrm{i}}$ defined by the initial value of coupling $g_\mathrm{i}$ to the instantaneous operators $\eta_{k,t}$ that are given by $g(t)$, then substitute Eq. (B.3) to account for the dynamics. The result can be simply expressed as the following scalar products:

$$a_k(t) = \begin{pmatrix} U_k & -V_k \end{pmatrix}\begin{pmatrix} u_k(t) \\ v^*_{-k}(t) \end{pmatrix}, \qquad b_k(t) = \begin{pmatrix} V_k & U_k \end{pmatrix}\begin{pmatrix} u_k(t) \\ v^*_{-k}(t) \end{pmatrix}, \tag{B.6}$$

where $U_k$ and $V_k$ are defined by Eq. (B.2) using the ramped coupling $g(t)$. The population of the mode $k$ is given by $n_k(t) = |b_k(t)|^2$. Notice that the slight difference between Eq. (B.6) and the notation of Refs. [26,33] is due to a different convention of the Bogoliubov transformation.

To take the continuum limit, one has to apply the prescriptions detailed in Sec. 2.2 to Eq. (B.4). Denoting the momentum of field theory modes with $p$ we get

$$A_p(t) = M(t), \qquad B_p = p, \tag{B.7}$$

where $M(t)$ is the time-dependent coupling of the field theory. The initial conditions read

$$u_p(t=0) = U_p, \qquad \left.\frac{\partial}{\partial t}u_p(t)\right|_{t=0} = -\iota M_\mathrm{i}U_p - \iota p V_{-p} \tag{B.8}$$

$$v^*_{-p}(t=0) = V_{-p}, \qquad \left.\frac{\partial}{\partial t}v^*_{-p}(t)\right|_{t=0} = -\iota p U_p + \iota M_\mathrm{i}V_{-p}, \tag{B.9}$$

where the adiabatic coefficients $U$ and $V$ are defined by the initial coupling $M_\mathrm{i}$ via the expressions

$$U_p = +\sqrt{\frac{1}{2} + \frac{M}{2\sqrt{p^2 + M^2}}} \tag{B.10}$$

and

$$V_p = \begin{cases} +\sqrt{\frac{1}{2} - \frac{M}{2\sqrt{p^2+M^2}}} & \text{for} \quad p \leq 0, \\ -\sqrt{\frac{1}{2} - \frac{M}{2\sqrt{p^2+M^2}}} & \text{for} \quad p > 0. \end{cases} \tag{B.11}$$

We remark that for a linear ramp profile one can express the solution exactly using the parabolic Weber functions [54]. However, for practical purposes we opted for the numerical

Table C.1: Matrix size vs. cutoff

| $N_{cut}$ | matrix size | $N_{cut}$ | matrix size | $N_{cut}$ | matrix size |
|---|---|---|---|---|---|
| 25 | 1330 | 35 | 9615 | 45 | 56867 |
| 27 | 1994 | 37 | 14045 | 47 | 78951 |
| 29 | 3023 | 39 | 20011 | 49 | 110053 |
| 31 | 4476 | 41 | 28624 | 51 | 151270 |
| 33 | 6654 | 43 | 40353 | 53 | 207809 |

integration of Eq. (B.4). The results of Sec. 3.1 are obtained by solving the differential equations substituting the quantised momenta for $p$. As the excitation probability of a mode $p$ is suppressed as $n_p \propto \exp(-\pi \tau_Q p^2 / m)$, we calculated the solution up to a momentum cut-off $p_{max}/m = 2\pi$. At volume $L = 50$ this amounts to 100 modes in the two sectors together.

For the intensive quantities considered in Secs. 4 and 5 we worked in the thermodynamic limit $L \to \infty$ where the sum over momentum modes is converted to an integral. Calculating the excitation probabilities of several modes up to a cutoff $p_{max}/m = 30$ we used interpolation to obtain a continuous $n_p$ function. This was used in the momentum integrals that yield the energy density and its higher cumulants. The need for the higher cutoff stems from the fact that $n_p$ is multiplied with higher powers of the dispersion relation for higher cumulants.

# C  TCSA: detailed description, extrapolation

## C.1  Conventions and applying truncation

The Truncated Conformal Space Approach was developed originally by Yurov and Zamolodchikov [65, 66]. It constructs the matrix elements of the Hamiltonian of a perturbed CFT in finite volume $L$ on the conformal basis. For the Ising Field Theory the critical point is described in terms of the $c = 1/2$ minimal CFT and adding one of its primary fields $\phi$ as a perturbation yields the dimensionless Hamiltonian:

$$H/\Delta = (H_0 + H_\phi)/\Delta = \frac{2\pi}{l} \left( L_0 + \bar{L}_0 - c/12 + \tilde{\kappa} \frac{l^{2-\Delta_\phi}}{(2\pi)^{1-\Delta_\phi}} M_\phi \right), \tag{C.1}$$

where $\Delta$ is the mass gap opened by the perturbation, $l = \Delta L$ the dimensionless volume parameter and $\Delta_\phi$ is the sum of left and right conformal weights of the primary field $\phi$. The matrix elements of $H$ are calculated using the eigenstates of the conformal Hamiltonian $H_0$ as basis vectors:

$$H_0 |n\rangle = \frac{2\pi}{L} \left( L_0 + \bar{L}_0 - \frac{c}{12} \right) |n\rangle = E_n |n\rangle, \tag{C.2}$$

where $c = 1/2$ is the central charge. The truncation is imposed by the constraint that only vectors with $E_n < E_{cut}$ are kept, where $E_{cut}$ is the cut-off energy. It is convenient to characterise the cut-off with the $L_0 + \bar{L}_0$ eigenvalue $N$ instead of the energy as it is related to the conformal descendant level. Table C.1 contains the number of states with

$$N - \frac{c}{12} < N_{cut} \equiv \frac{L}{2\pi} E_{cut} \tag{C.3}$$

for the range of cut-offs that were used in this work. We remark that the maximal conformal descendant level $\mathcal{N}_{max}$ is related to the cut-off parameter as $\mathcal{N}_{max} = (N_{cut} - 1)/2$.

Table C.2: Extrapolation exponents

| Observable | Free fermion model | | $E_8$ model | |
| --- | --- | --- | --- | --- |
| | Leading | Subleading | Leading | Subleading |
| $\kappa_n$ | -1 | -2 | -11/4 | -15/4 |
| $\sigma$ | -1 | -2 | -7/4 | -11/4 |
| Overlap | -1 | -2 | -11/4 | -15/4 |

## C.2 Extrapolation details

To reduce the truncation effects, we employ the cut-off extrapolation scheme developed in Ref. [75]. A detailed description of this scheme is presented in Ref. [81], here we merely discuss its application to the quantities considered in the main text. For some observable $\mathcal{O}$ the dependence on the cut-off parameter $N_{\text{cut}}$ is expressed as a power-law:

$$\langle \mathcal{O} \rangle = \langle \mathcal{O} \rangle_{\text{TCSA}} + A N_{\text{cut}}^{-\alpha_{\mathcal{O}}} + B N_{\text{cut}}^{-\beta_{\mathcal{O}}} + \dots . \tag{C.4}$$

The exponents $\alpha < \beta$ depend on the observable $\mathcal{O}$, the operator that perturbs the CFT, and on those entering the operator product expansion of the above two. For the excess energy and the magnetisation one-point function as well as the overlaps it is straightforward to apply this recipe to obtain the leading and subleading exponents. In the case of higher cumulants of the excess heat there is no existing formula. However, as they can be expressed as the sum of products of energy levels and overlaps, the leading and subleading exponents coincide with those of the first cumulant, i.e. the excess heat. The exponents are summarised in Table C.2. Sampling the dynamics using different cut-off parameters we obtained the extrapolated results by fitting the expression Eq. (C.4) to our data. In certain cases the fit with two exponents proved to be numerically unstable reflected by large residual error of the estimated fit coefficients. In these cases, only the leading exponent was used. For dynamical one-point functions the extrapolation procedure was applied in each "time slice". As evident from the exponents, the $E_8$ model exhibits faster convergence in terms of the cut-off. However, in most of the cases the extrapolation scheme yields satisfactory results in the FF model as well, with the notable exception of the magnetisation, as discussed in the main text. Let us now present how the extrapolation works for various quantities to illustrate its preciseness and limitations.

Let us start with calculations concerning dynamics on the free fermion line. Out of the two dynamical one-point functions, the order parameter is more sensitive to the TCSA cut-off. Fig. C.1. presents an example of the cut-off extrapolation for this quantity with $M_i L = 50$ and $M_i \tau_Q = 128$. The extrapolation error (denoted by a grey band around the curve) is relatively large and partly explains the lack of dynamical scaling before the impulse regime in Fig. 4.1b. We remark that in this case the two-exponent fit was unstable hence only the leading term of Eq. (C.4) was used. The dependence on the cut-off is less drastic for shorter ramps.

The energy density exhibits much faster convergence in terms of cut-off in both models. It is in fact invisible on the scale of Figs. 4.1a and 4.2a, consequently we do not present the details of their extrapolation here. To make contrast with Fig. C.1, we illustrate with Fig. C.2 that the time evolution of the magnetisation operator is captured much more accurately by TCSA in the $E_8$ model. The two-exponent fit is numerically stable in this case hence we use both the leading and the subleading exponent to determine the infinite cut-off result. The change between data obtained using different cut-off parameters and the extrapolation error falls within the range of the line width in almost the whole duration of the ramp.

Apart from dynamical expectation values of local observables, we also discussed higher cumulants of work in the main text. Although the use of TCSA to directly calculate such quantities is unprecedented, based on the discussion following Eq. (C.4) we expect that the same

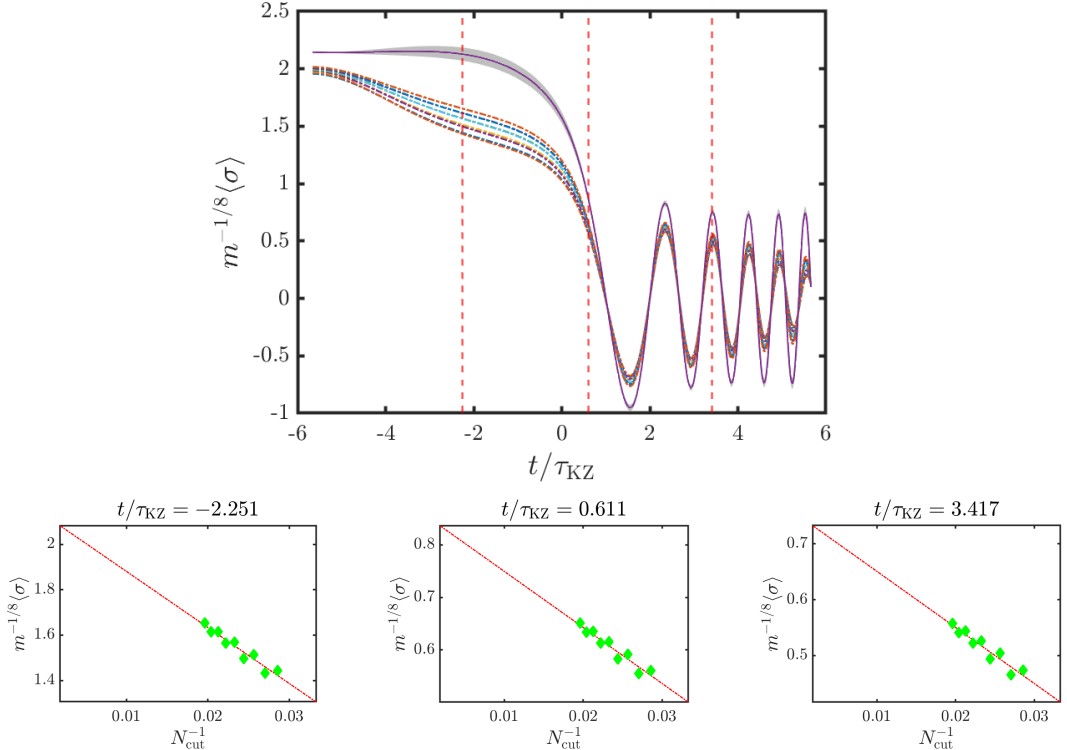

Figure C.1: Details of the extrapolation for the dynamical one-point function of the order parameter for a ferromagnetic-paramagnetic ramp along the free fermion line with $mL = 50$ and $m\tau_Q = 128$. Raw TCSA data are plotted in dot-dashed lines in the main figures, the cut-off parameter is in the range $N_{cut} = 35 \dots 51$. Extrapolated data is denoted by solid lines, with the residual error as a grey shading. Dashed red lines correspond to the time instants that are detailed in the subplots. Green diamonds denote raw data as a function of $N_{cut}^{-1}$ where $-1$ is the leading exponent. Red dashed lines denote the fitted function.

expression accounts for the cut-off dependence as in the case of local observables. This is what we find inspecting Fig. C.3. The depicted data is a small subset of all the extrapolations whose results are presented in the main text but they convey the general message that cumulants can be obtained accurately using TCSA. The relative error in the extrapolated value is typically in the order of $1-3\%$ for cumulants in the free fermion model (with an increase towards higher cumulants) and around $0.1-0.7\%$ in the $E_8$ model.

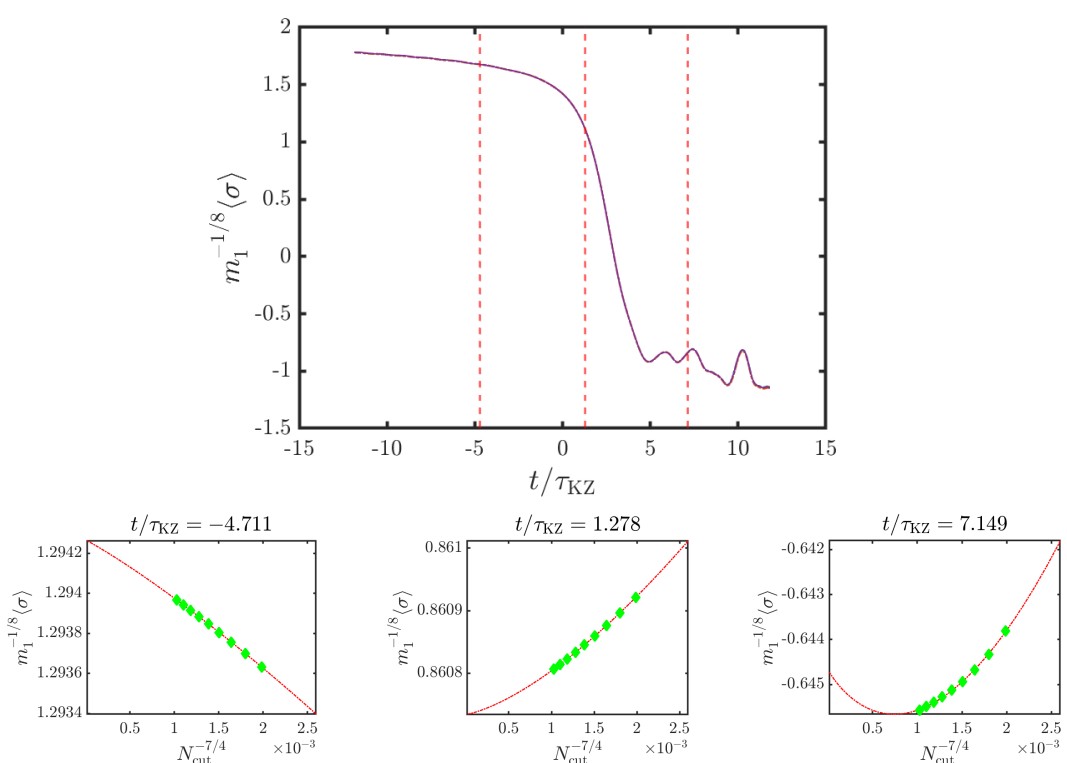

Figure C.2: Details of the extrapolation for the dynamical one-point function of the magnetisation ramp along the $E_8$ line with $m_1 L = 50$ and $m_1 \tau_Q = 128$. Notations and range of cut-offs is the same as in Fig. C.1. Note the range of the $y$ axis in the subplots.

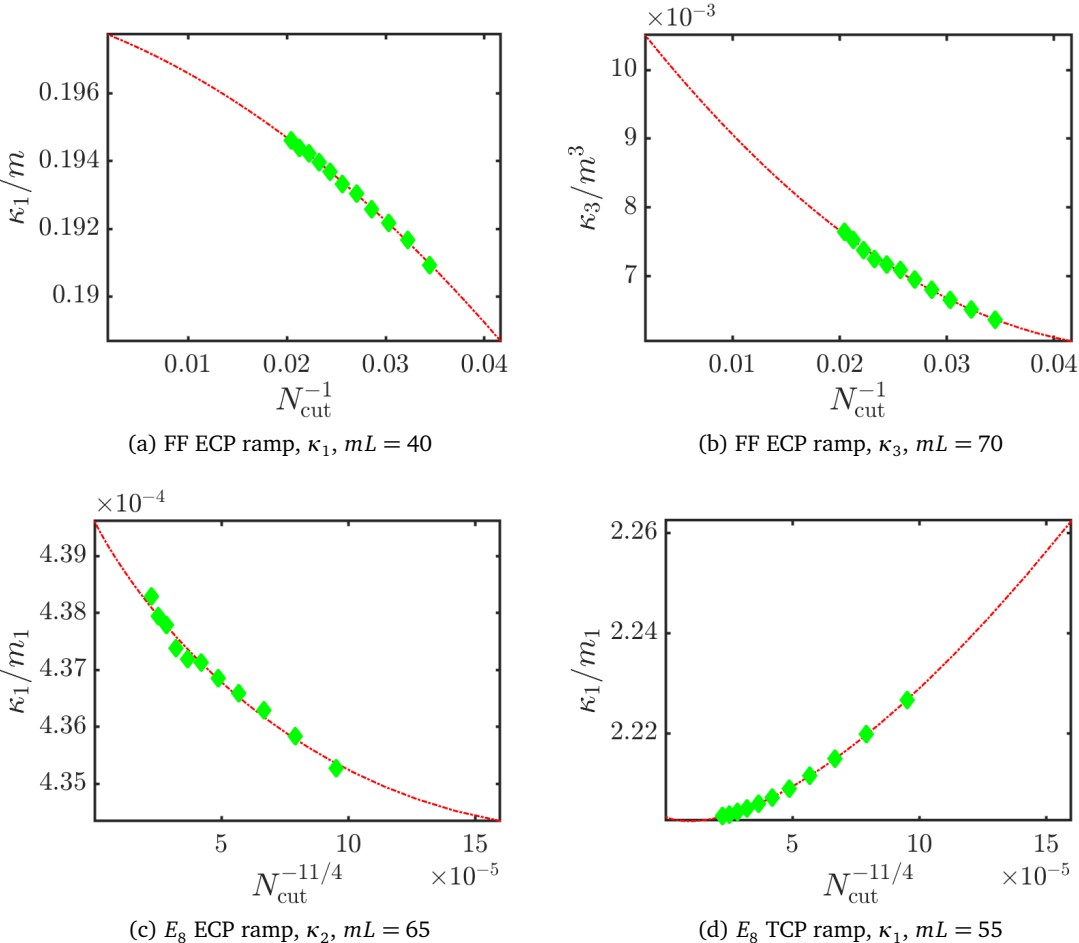

(a) FF ECP ramp, $\kappa_1$, $mL = 40$

(b) FF ECP ramp, $\kappa_3$, $mL = 70$

(c) $E_8$ ECP ramp, $\kappa_2$, $mL = 65$

(d) $E_8$ TCP ramp, $\kappa_1$, $mL = 55$

Figure C.3: Extrapolation of various work cumulants for various protocols. The plots are typical of the overall picture of extrapolating overlaps obtained using TCSA.

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
