# Peer review of "Kibble–Zurek mechanism in the Ising Field Theory"

_SciPost Physics, doi:SciPost Phys. 9, 055 (2020)_

## Round 1 · Referee Report · Anonymous · 2020-8-3

Strengths
The authors perform an in-depth study of the quantum Kibble-Zurek problem for the Ising Field Theory. They analyze a vast zoology of regimes and quench directions in the 2d coupling space, by employing advanced analytical and numerical techniques inspired by conformal field theory. The approach is well justified, and the results are well argumented and scientifically sound.
Weaknesses
1. The delivery of take-away messages is somewhat mediocre: The most interesting results of the research are scattered throughout the text, partially hidden in the conclusion, and almost absent in the abstract.
2. The language is slightly too focused on statistical quantum mechanics: Even though the potential audience for this research may be broad, it feels that the adopted language relies on conformal field theory language more than it should.
3. Proposal of experiments missing: There is no mention how the additional knowledge made available by this study may impact or be verified on an experimental platform.
Report
I will likely recommend publication after the authors consider my comments and suggestions.
Requested changes
To overcome the weaknesses I previously mentioned, I would suggest the following revision (using the same numbering as before):
1. Highlight the most important results better, in the abstract AND the conclusion.
2. Expand the heavy-CFT oriented arguments by making them more approachable for non-experts.
3. Consider if the results from this research can foster the design of new experiments, or can be verified in existing experiments, or if they explain previously-observed unsolved phenomena.
Author: Márton Kormos on 2020-09-11 [id 959]
(in reply to Report 1 on 2020-08-03)
We thank the Referee for reading our manuscript and for their useful remarks. We are glad that the Referee finds our work well argumented and scientifically sound. We found their observations very useful and in response to them, we implemented the following changes:
-
In order to highlight our key results more clearly, we expanded the abstract and modified the Conclusions.
-
To avoid unnecessary or unclear references to CFT, we expanded and partially rewrote Sec. 2.2 that outlines the model. Instead of referring to the CFT, we use the expressions "quantum critical point" and "free massless Majorana theory" which are much more accessible to a broader audience. Some CFT concepts lie at the heart of our numerical method and thus are indispensable. The remaining appearances of CFT and integrable field theory concepts support the internal coherence of the text, articulating the field theoretical nature of our work.
-
We added a paragraph to the Conclusions discussing possible experiments with cold atomic gases and Rydberg atoms in which our predictions could be investigated.
We believe that the weaknesses pointed out by the Referee are eliminated by the refined presentation.
Author: Márton Kormos on 2020-09-11 [id 961]
(in reply to Report 3 by Dirk Schuricht on 2020-08-29)We are grateful to the Referee for the careful reading of the manuscript and his valuable feedback. We are pleased by his opinion that our paper presents a systematic analysis in a very clear way.
Let us address his questions and comments in order.
and 3.: Indeed, our ramp definition was not fully precise. Now in Sec. 2.2 we describe the various quench protocols more precisely, eliminating previous inconsistencies.
We included the masses of the single-particle states of the $E_8$ model and commented on its exact S-matrix, citing the relevant references.
This mismatch is absent from the integrated quantities like the overall defect density, the excess heat density, and their respective cumulants. A remote analogy can be drawn with the form factor series expansion calculation of the central charge in integrable perturbed conformal field theories, where the result of the sum over multiparticle states is fixed by the $c$-theorem, while the separate terms vary greatly due to the details of the interaction. To improve our presentation, we merged the two concluding paragraphs of Sec. 2.3.1 with the concluding paragraph of Sec. 3.2, where we discuss the applicability of the APT in light of the above discrepancy.
The reasoning leading to Eq. (4.3) assumes that due to the critical slowing down the only relevant time scale is set by the "freeze-out" time. The domain of validity for this assumption is in the "impulse" regime, i.e. for -$\tau_\text{KZ} < t < \tau_\text{KZ}$ (see in detail in Ref. [31]). Hence, in general we do not expect Eq. (4.3) to apply for the late-time dynamics, with a notable exception given by the energy density, as discussed around Eq. (4.4).
We believe that the above modifications, based on the Referee's advice, have substantially improved the manuscript.

---

## Round 1 · Referee Report · Anonymous · 2020-8-5

Report
The manuscript provides solution of the Kibble-Zurek problem in a genuinely interacting quantum field theory. This is a valuable extension that may open a new research avenue. The calculation involves approximations, naturally. The crucial one is the adiabatic perturbation theory (APT) whose validity is claimed to be justified for low density of excitations. However, comparison between the APT in Ref. [7] and an exact solution in Ref. [8] shows that in the transverse Ising chain the APT gives the correct power law for density of defects but its numerical prefactor is overestimated by 60%. This error occurs for low density of excitations that are obtained for long quench times and, therefore, the low density criterion is not sufficient to justify the APT. I would like the authors to make more thorough analysis of APT's applicability before the manuscript can be accepted for publication.
Author: Márton Kormos on 2020-09-11 [id 960]
(in reply to Report 2 on 2020-08-05)
We thank the Referee for their question which allows us to further clarify this issue.
First, we would like to stress that we did not attempt to extract numerically accurate values using adiabatic perturbation theory (APT), but we used the formalism to support the scaling laws. However, the Referee's question is of course relevant, as the validity of the APT would be questioned by a huge numerical deviation from the exact results.
So let us address the 60% discrepancy mentioned by the Referee. We believe it probably comes from a typo in Ref. [24] (ref. number in our resubmitted manuscript), namely in Eq. (4.89) (or Eq. (89) in the arXiv version). However, comparing Eqs. (4.26), (4.88) and (4.28), we find
\[ |\alpha_k|^2 = \pi^2/9 e^{-2\pi k^2/\delta}. \]
Then integrating with respect to $k$ as in Eq. (4.89) one obtains
\[ \pi^2/9 \cdot 1/(2\pi) \sqrt{\delta/2} = 0.1234... \sqrt{\delta}. \]
This essentially agrees with what is written in the review [26] by Dziarmaga below Eq. (121). We suspect that the first two digits in Eq. (4.89) of Ref. [24] got swapped accidentally. We note that Ref. [24] presents the most detailed derivation of the APT kink density after the ramp. In particular, it is more accurate than the estimate in Ref. [7]. While the numerical values of the prefactor differ in Refs. [7, 24, 26], a careful examination of the details in Ref. [24] show that the discrepancy with respect to the exact prefactor $1/2\sqrt{2}\approx0.1125$ is given by the factor $\pi^2/9$, which amounts to a discrepancy around 10% instead of 60%.
As discussed in Ref. [24], the reason why APT overestimates the density of defects is that the first order approximation does not take into account the fermionic nature of the particles which prevents them from occupying the same momentum state. (For bosons, the APT result underestimates the true value.) This is manifest in that $|\alpha_k|^2 > 1$ is permitted by the APT despite the fact that it represents a fermionic occupation number of the mode with momentum $k$. However, even if the numerical prefactor has some error, the scaling dependence on the ramp velocity is correctly captured and this is what we exploit.
Finally, let us emphasise that while we use APT to facilitate some of the scaling arguments, we always put these scaling laws to numerical test as the core of our work is of numerical nature.
Anonymous on 2020-09-17 [id 976]
(in reply to Márton Kormos on 2020-09-11 [id 960])The reply makes sense. No further comments. The manuscript is ready for publication.

---

## Round 1 · Referee Report · Dirk Schuricht · 2020-8-29

Strengths
1. Systematic analysis of the scaling behaviour of several quantities in the Ising field theory.
2. Good overview of expected scaling behaviour.
3. Good discussion and outlook.
Weaknesses
1. The definition of the ramp protocols is not consistent. For example, it is unclear how to interpret (2.1) for negative times which are used later.
2. Discussion of the properties of the E_8 model, eg, the values of the other masses along with (2.14), could be included.
3. Discussion around (2.15) imprecise. For example, the initial and final values of \lambda are \mp\lambda_0, so it is unclear how a ramp to the critical point can be achieved.
4. At the end of Sec. 3.2 it is discussed that the results shown in Fig. 3.5 do not agree with the predictions from perturbation theory, however, no explanation for this discrepancy is given.
5. The scaling collapse seems not to be working for the magnetisation in Fig. 4.2(b). Is there an understanding why? As far as I see the argument leading to (4.3) with different exponents should still apply, right?
Report
The authors study the Kibble-Zurek physics for linear ramps in the Ising field theory. This offers two different paths for the ramp through the critical point: a non-interacting and an interacting one. The two directions have very different scaling laws. The method to study both is the truncated conformal space approach, while for the first case also an exact solution can be obtained. The authors observe scaling behaviour in several quantities in agreement with general scaling arguments. The paper is nicely written and the presentation very clear. There are a few minor remarks (see below) which should be addressed, once this has been done I recommend publication.
Requested changes
See weaknesses.

---

## Round 2 · Referee Report · Anonymous (Referee 1) · 2020-9-17

Report

The authors have addressed the issues I raised, and I am satisfied with the changes. I recommend the new version for publication.

---

## Round 2 · Referee Report · Dirk Schuricht (Referee 3) · 2020-9-21

Report

The authors have properly addressed all points. The manuscript can be accepted.

---

## Round 2 · Author Response

Dear Editor,

The three referee reports were quite positive, and based on them we were asked to perform minor revisions of our manuscript. All the three reports raised valid and interesting points. We think we gave detailed answers to their questions, and we believe that our paper has improved considerably by the modifications done based on their comments and questions. We hope that our manuscript is now suitable for publication.

Yours sincerely,
Kristóf Hódsági and Márton Kormos

---

## Round 2 · List of Changes

• In order to highlight our key results more clearly, we expanded the Abstract and modified the Conclusions.

  • To avoid unnecessary or unclear references to CFT, we expanded and partially rewrote Sec. 2.2 that outlines the model. Fig. 2.1 has also changed slightly.

  • In Sec. 2.2 now we describe the various quench protocols more precisely, eliminating previous inconsistencies.

  • We included the masses of the single-particle states of the $E_8$ model (see Eq. (2.15)) and commented on its exact S-matrix, citing the relevant references.

  • We merged the two concluding paragraphs of Sec. 2.3.1 with the concluding paragraph of Sec. 3.2.

  • We added a paragraph to the Conclusions discussing possible experiments with cold atomic gases and Rydberg atoms in which our predictions could be investigated.

---

## Editorial Decision

published